# Palmitoylation prevents B7-H4 lysosomal degradation sustaining tumor immune evasion

Yijian Yan[1,2,7], Jiali Yu[1,2,7], Weichao Wang[1,2], Ying Xu[1,2], Kole Tison[1,2], Rongxin Xiao[1,2], Sara Grove[1,2], Shuang Wei[1,2], Linda Vatan[1,2], Max Wicha[3], Ilona Kryczek[1,2] & Weiping Zou[1,2,4,5,6] ✉

B7-H4 functions as an immune checkpoint in the tumor microenvironment (TME). However, the post-translational modification (PTM) of B7-H4 and its translational potential in cancer remains incompletely understood. We find that ZDHHC3, a zinc finger DHHC-type palmitoyltransferase, palmitoylates B7-H4 at Cys130 in breast cancer cells, preventing its lysosomal degradation and sustaining B7-H4-mediated immunosuppression. Knockdown of ZDHHC3 in tumors results in robust anti-tumor immunity and reduces tumor progression in murine models. Moreover, abemaciclib, a CDK4/6 inhibitor, primes lysosome activation and promotes lysosomal degradation of B7-H4 independently of the tumor cell cycle. Treatment with abemaciclib results in T cell activation and mitigates B7-H4-mediated immune suppression via inducing B7-H4 degradation in preclinical tumor models. Thus, B7-H4 palmitoylation is an important PTM controlling B7-H4 protein stability and abemaciclib may be repurposed to promote B7-H4 degradation, thereby treating patients with B7-H4 expressing tumors.

Immune checkpoint blockade (ICB) therapy, targeting PD-1/PD-L1 (B7-H1) and CTLA-4, has revolutionized cancer treatment[1,2]. B7-H4 (also known as VTCN1, B7S1, and B7x) is a single transmembrane glycoprotein belonging to the B7 family. B7-H4 inhibits T cell function through unidentified receptors[3–7]. High levels of B7-H4 expression are observed in various cancer types, including breast, ovarian, and uterine cancers[7–11]. Additionally, B7-H4 and PD-L1 often exhibit mutually exclusive expression in cancer cells[7,10,12,13]. As a result, B7-H4 represents an appealing therapeutic target. Despite substantial efforts on developing specific inhibitors or blocking antibodies against B7-H4, yet there is no FDA-approved drug currently targeting B7-H4.

Since the receptors for B7-H4 remain to be identified, it is even more critical to dissect the molecular mechanisms controlling B7-H4

mRNA expression and protein stability. Early studies have largely focused on the transcriptional regulation of B7-H4, rather than post-translational modification (PTM)[14–17]. It appears that B7-H4 transcripts are expressed in many tissues, while B7-H4 protein is expressed in specific cancer types and limited tissues[18]. One recent report shows that N-linked glycosylation shields B7-H4 in breast cancer cells from ubiquitination by the E3 ligase autocrine motility factor receptor (AMFR), preventing subsequent proteasomal degradation[7]. However, it remains unclear whether other types of PTM master the turnover of B7-H4 in different tumor types.

In this work, we conduct a drug screen targeting various PTMs in B7-H4 expressing breast cancer cells. We discover that B7-H4 is palmitoylated via ZDHHC3 at Cys130 and B7-H4 palmitoylation protects it

[1]Department of Surgery, University of Michigan Medical School, Ann Arbor, MI, USA. [2]Center of Excellence for Cancer Immunology and Immunotherapy, University of Michigan Rogel Cancer Center, Ann Arbor, MI, USA. [3]Department of Internal Medicine, University of Michigan Medical School, Ann Arbor, MI, USA. [4]Department of Pathology, University of Michigan Medical School, Ann Arbor, MI, USA. [5]Graduate Program in Immunology, University of Michigan, Ann Arbor, MI, USA. [6]Graduate Program in Cancer Biology, University of Michigan, Ann Arbor, MI, USA. [7]These authors contributed equally: Yijian Yan, Jiali Yu. ✉e-mail: wzou@umich.edu

from lysosomal degradation. Based on the insights into B7-H4 degradation via lysosomes, we find that abemaciclib, an FDA-approved CDK4/6 inhibitor for breast cancer, reduces B7-H4 protein levels by enhancing lysosome biogenesis and abundance. We further demonstrate that abemaciclib treatment inhibits tumor growth by enhancing B7-H4 degradation in vivo and counteracting B7-H4-mediated immune suppression.

## Results

### B7-H4 in breast cancer cells promotes tumor progression

B7-H4 is highly expressed in various types of cancer, including breast cancer[7–11]. Despite its reported immune inhibitory role[10], the immunological effect of endogenous B7-H4 in tumors remains unclear. This discrepancy is partly attributed to the lack of well-established tumor models with endogenous B7-H4 expression in syngeneic immune-competent mice. To address the question, we established an orthotopic breast cancer cell line, 4H11, originating from tumors induced by medroxyprogesterone acetate (MPA) plus 7,12-dimethylbenz[a]anthracene (DMBA)[19,20] (Supplementary Fig. 1a). We detected strong endogenous B7-H4 expression in 4H11. Then, we established B7-H4 knockout (KO) 4H11 cells (Supplementary Fig. 1b). B7-H4$^{+/+}$ (WT) and B7-H4$^{-/-}$ (KO) 4H11 cells comparably proliferated in vitro (Supplementary Fig. 1c). Then, we inoculated WT and KO 4H11 cells into wild type C57BL/6 mice. B7-H4 deficiency significantly suppressed tumor growth in the immune-competent mice (Fig. 1a, b), but not in NOD.SCID gc deficient (NSG) mice (Supplementary Fig. 1d). Consistent with this observation, we detected more tumor-infiltrating CD8$^+$ T cells with increased TNF-α production in the B7-H4-KO tumors compared to the WT tumors (Fig. 1c and Supplementary Fig. 1e). T cells recovered from B7-H4-KO tumors exhibited a reduced exhaustion phenotype (PD-1$^+$TIM-3$^+$) compared to T cells from WT tumors (Fig. 1d). Furthermore, there were more stem-like TCF-1$^+$CD8$^+$T cells and less exhausted TOX$^+$CD8$^+$ T cells in B7-H4-KO tumor tissues (Fig. 1e–g). These data demonstrate an immunosuppressive role of endogenous B7-H4 in mouse breast cancer.

We further validated the immunosuppressive function of B7-H4 in 4T1 breast cancer model. As 4T1 cells expressed undetectable endogenous B7-H4, we overexpressed B7-H4 in 4T1 cells (B7-H4-OE) (Supplementary Fig. 1f). Then, we inoculated 4T1 control (containing empty vectors, EV) and B7-H4-OE tumors into BALB/c mice. B7-H4-OE tumors exhibited faster progression than control tumors, as shown by tumor volume and weight (Fig. 1h–j). Exogenous expression of B7-H4 in 4T1 resulted in fewer IFN-γ$^+$ and TNF-α$^+$CD8$^+$ T cells in the tumor microenvironment as compared to controls (Fig. 1k–m). To determine the role of CD8$^+$ T cells in B7-H4-mediated immunosuppression, we depleted CD8$^+$ T-cells with anti-CD8 monoclonal antibody (mAb) in the 4T1 tumor model. We observed that CD8$^+$ T cell depletion led to an increase in tumor weight in 4T1 tumors, while the effect was compromised in mice bearing B7-H4-OE tumors (Fig. 1n). Collectively, we provide direct evidence showing an immunosuppressive role of B7-H4 in vivo.

### B7-H4 palmitoylation prevents its lysosomal degradation

PTMs regulate protein stability and subcellular localization[21,22]. To explore whether PTMs alter B7-H4 protein stability, we conducted a screen by pharmacologically blocking different types of PTM in human breast MDA-MB-468 cancer cells. To this end, we created MDA-MB-468 cells expressing B7-H4-GFP under the control of the CMV promoter (Supplementary Fig. 2a). We used flow cytometry to assess the GFP signal intensity in MDA-MB-468 cells treated with different PTM inhibitors in the absence or presence of proteasome/lysosome inhibitors. We found that tunicamycin, a glycosylation inhibitor, reduced the B7-H4 protein level, which was rescued by MG132, a proteasome inhibitor[7] (Fig. 2a). Interestingly, 2-Bromopalmitate (2-BP), a palmitoylation inhibitor, also profoundly reduced the B7-H4 protein level. This effect

was largely reversed by bafilomycin A1, a lysosome inhibitor, but not by MG132 (Fig. 2a), suggesting the lysosome- but not proteasome-dependent effect of 2-BP. We examined the effect of 2-BP on endogenous B7-H4 in human (MDA-MB-468 and T-47D) and mouse (4H11, Py230, and MMTV-PyMT) breast cancer cells. Treatment with 2-BP led to reduced B7-H4 expression in these cells in a dose-dependent manner (Fig. 2b–e and Supplementary Fig. 2b). 2-BP is a protein palmitoylation inhibitor. The data suggest that palmitoylation may result in B7-H4 protein stabilization. To further investigate this, we treated cells with palmostatin B, an inhibitor of the depalmitoylation enzyme APT1/2[23,24]. We observed an increase in B7-H4 protein in these breast cancer cells (Fig. 2b–d and Supplementary Fig. 2c). Treatment with cerulenin, another palmitoylation inhibitor[25,26], also reduced B7-H4 protein in MDA-MB-468, PyMT, 4H11, and Py230 cells (Fig. 2f and Supplementary Fig. 2d–f). Thus, palmitoylation can stabilize B7-H4 protein expression in multiple mouse and human breast cancer cells.

To validate whether B7-H4 is palmitoylated, we applied the biotin acyl-exchange (ABE) method to detect the palmitoylation of endogenous B7-H4 in breast cancer cells. We detected B7-H4 palmitoylation in human (MDA-MB-468 and T47D) and mouse (4H11 and Py230) breast cancer cells. As expected, treatment with 2-BP reversed B7-H4 palmitoylation (Fig. 2g, h and Supplementary Fig. 2g, h). Using the Click-iT assay, we further validated the occurrence of B7-H4 palmitoylation in MDA-MB-468 cells (Supplementary Fig. 2i). We extended our investigation to additional cancer and tissue. We detected B7-H4 palmitoylation in both normal mouse uterus and human ovarian cancer tissues (Supplementary Fig. 2j, k). Together, these findings indicate that B7-H4 is subjected to palmitoylation in breast and ovarian cancer cells as well as normal uterine tissues.

We next tested how effective palmitoylation regulates B7-H4 expression. Cycloheximide (CHX) chase analysis revealed that 2-BP treatment resulted in accelerated B7-H4 reduction compared to CHX alone in both human and mouse breast cancer cells (Fig. 2i, j and Supplementary Fig. 2l). To gain mechanistic insights into how palmitoylation stabilizes B7-H4 protein, we employed protein degradation inhibitors to block either proteasome- or lysosome-mediated degradation pathways alone or in the presence of 2-BP. We discovered that lysosome inhibitor bafilomycin A1 or the lysosome digestive enzyme inhibitors E64d plus pepstatin A, but not the proteasome inhibitor MG132 (Fig. 2a), rescued B7-H4 expression in MDA-MB-468, T-47D, and 4H11 cells (Fig. 2k–m). The results suggest that 2-BP treatment primes the lysosomal degradation of B7-H4. It is thought that cellular membrane B7-H4 mediates T cell inhibition[4]. We postulated that B7-H4 palmitoylation diminished its lysosomal degradation, resulting in an increase in cellular membrane B7-H4 expression. To test the hypothesis, we determined the cell surface B7-H4 with or without 2-BP treatment in MDA-MB-468 cells. 2-BP abrogated the total and cell surface B7-H4 (Supplementary Fig. 2m). Interestingly, the addition of bafilomycin A1 recovered the total B7-H4, but not cell membrane B7-H4 (Supplementary Fig. 2m). The data suggest that palmitoylation deficient B7-H4 may be directed to the lysosome for degradation and can be stalled by bafilomycin A1 in the lysosome. Thus, B7-H4 is stabilized and protected by palmitoylation from lysosome-mediated degradation.

### ZDHHC3 catalyzes B7-H4 palmitoylation at Cys130 impairing tumor immunity

Palmitoylation occurs at cysteine residue[27]. To identify the palmitoylation site of B7-H4, we used mass spectrometry (MS) to detect the palmitoylation of B7-H4. We discovered that the palmitoyl group was located on the Cys130 residue (Supplementary Fig. 3a). Then, we created a B7-H4 mutant (C130A) by replacing cysteine with alanine. C130A mutation reduced B7-H4 palmitoylation levels as compared to WT (Fig. 3a). To explore additional potential palmitoylation sites, we mutated all cysteine residues to alanine and evaluated the palmitoylation

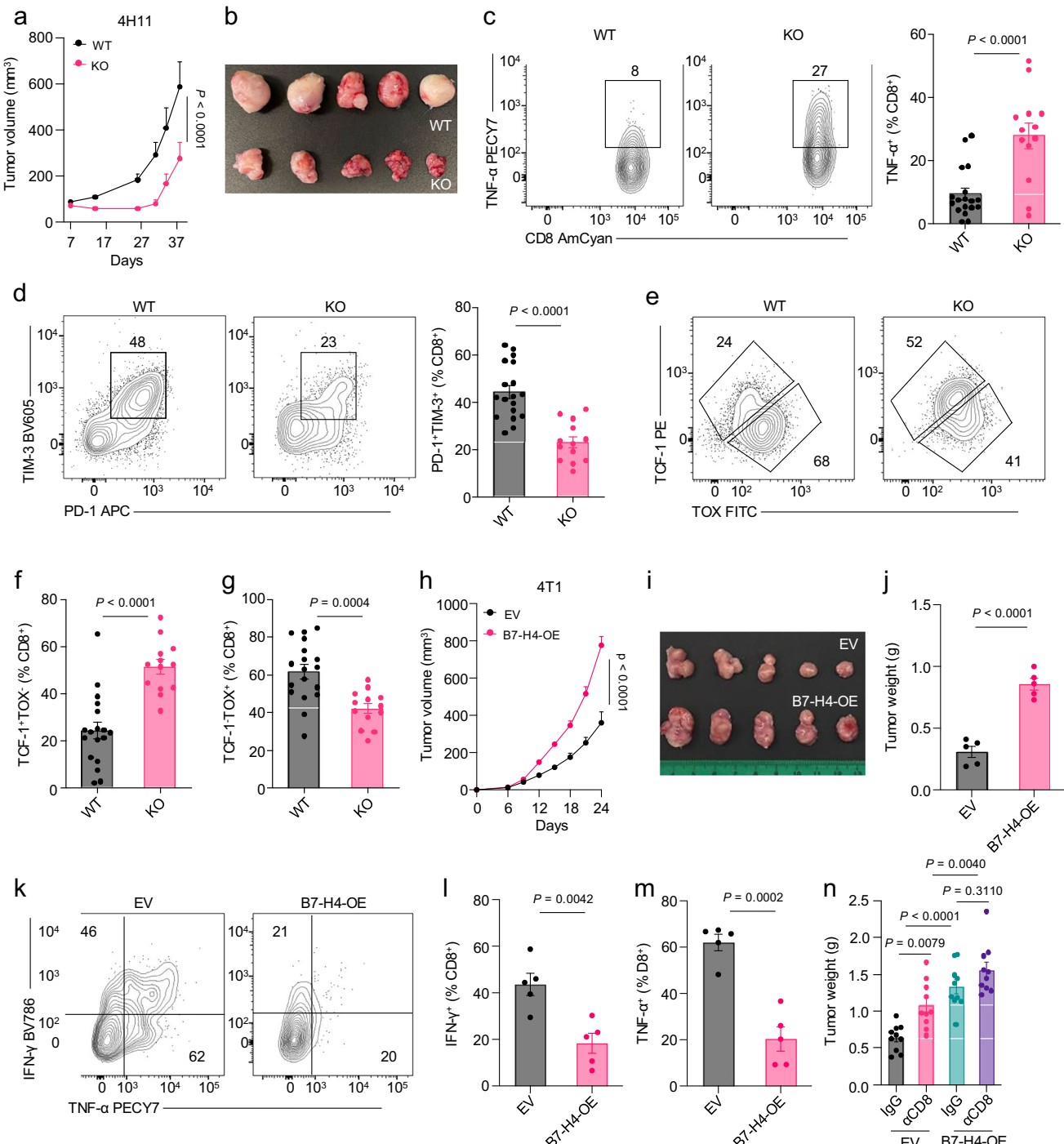

**Fig. 1 | B7-H4 in breast cancer cells promotes tumor progression.** B7-H4 knockout (KO) and wild-type (WT) 4H11 subcutaneous tumor growth (**a**) and representative image (**b**). n = 5 per group. Subcutaneous implantation was conducted in 10-week-old female C57BL/6 mice. Data are shown as mean ± SEM (**a**). Statistical analysis was performed using two-way ANOVA (**a**). Flow cytometry analysis of tumor-infiltrating TNF-α⁺ (**c**), TIM-3⁺PD-1⁺ (**d**), TCF-1⁺TOX⁻ (**e** and **f**), TCF-1⁻TOX⁺ (**e** and **g**) CD8⁺ T cells in subcutaneous 4H11 WT (n = 18) and B7-H4-KO (n = 13) tumor-bearing mice. Pooled data from 2 independent experiments. Subcutaneous implantation was conducted in 10-week-old female C57BL/6 mice. Data are shown as mean ± SEM. Statistical analyses were performed using two-tailed Student's *t* test (**c**, **d**, **f**, **g**). B7-H4 overexpression (B7-H4-OE) and empty vector (EV) harboring 4T1 subcutaneous tumor growth (**h**), representative image (**i**), and tumor weights (**j**). n = 5 per group. Subcutaneous implantation was conducted in 8-week-old female BALB/c mice. Data are shown as mean ± SEM (**h**, **j**). Statistical analyses were performed using two-way ANOVA (**h**) and two-tailed Student's *t* test (**j**). Flow cytometry analysis of tumor-infiltrating IFN-γ⁺ (**k** and **l**) and TNF-α⁺ (**k**, **m**) CD8⁺ T cells in 4T1 tumor-bearing BALB/c mice. n = 5 per group. Subcutaneous implantation was conducted in 8-week-old female BALB/c mice. Data are shown as mean ± SEM (**l**, **m**). Statistical analyses were performed using two-tailed Student's *t* test (**l**, **m**). **n** Tumor weights of B7-H4-OE and EV 4T1 tumors in mice treated with isotype control (IgG) or anti-CD8 mAb. n = 10 per group. Subcutaneous implantation was conducted in 8-week-old female BALB/c mice. Data are shown as mean ± SEM. Statistical analysis was performed using one-way ANOVA with Tukey's multiple comparison test. Source data are provided as Source Data File.

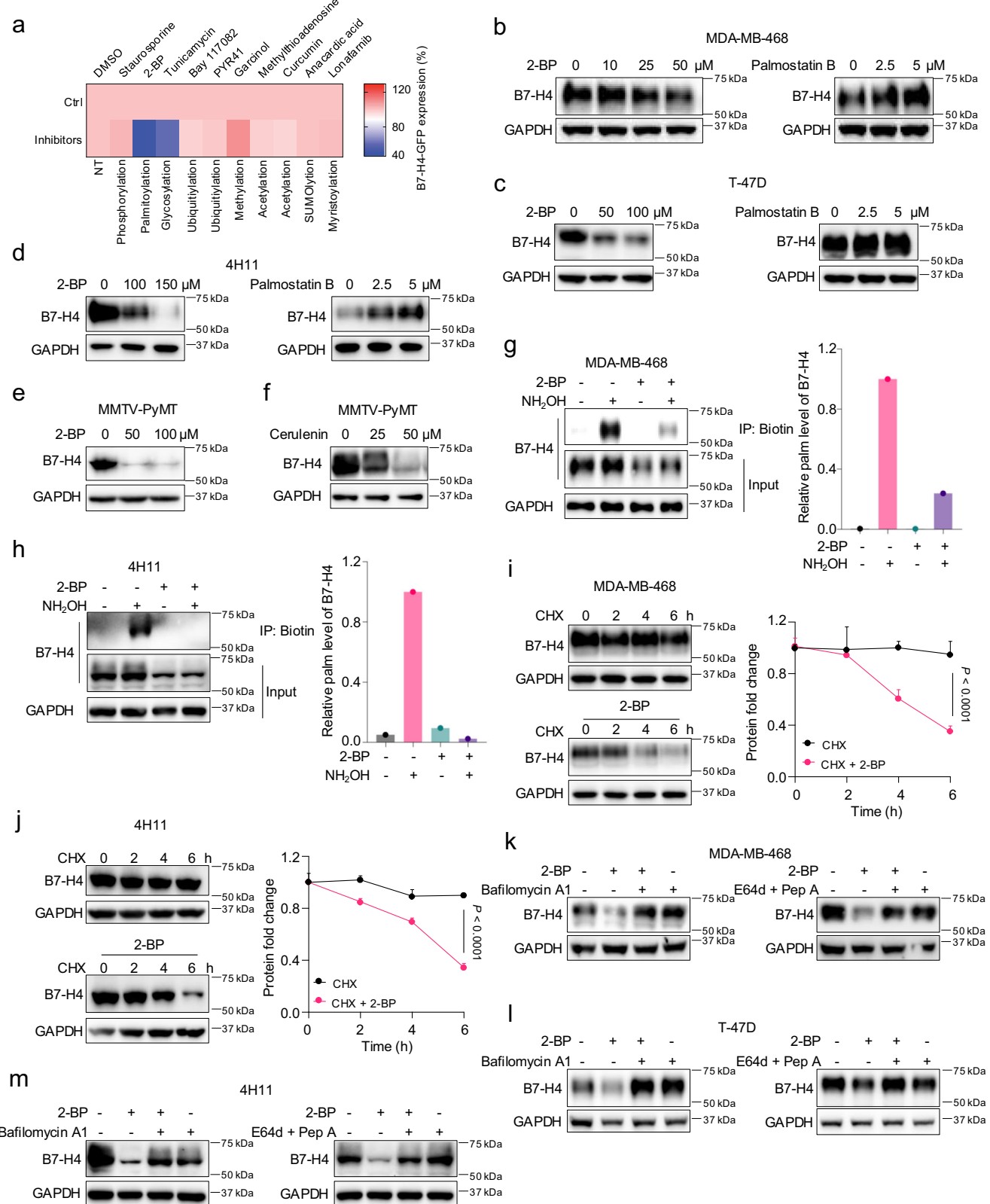

levels of these B7-H4 mutants. The ABE assay showed that mutations in the other cysteine residues had no effect on B7-H4 palmitoylation (Supplementary Fig. 3b). Given that palmitoylation sustains the protein stability of B7-H4 and 2-BP treatment results in lysosomal degradation of B7-H4, we examined the effect of 2-BP on B7-H4 expression in 293T cells expressing WT or C130A B7-H4. 293T cells do not express endogenous B7-H4. As expected, treatment with 2-BP abolished WT B7-H4, but not C130A mutant B7-H4 (Fig. 3b). Moreover, C130A B7-H4 manifested a faster rate of reduction in the CHX chase assay compared to WT B7-H4 (Fig. 3c). These experiments demonstrate that palmitoylation at C130 is essential for maintaining B7-H4 stability.

Palmitoylation constitutes a reversible lipidation process wherein the addition of a palmitoyl group to free cysteine residues is facilitated by palmitoyltransferase (PAT). Humans possess 23 PATs that are

**Fig. 2 | B7-H4 palmitoylation prevents its lysosomal degradation. a** Normalized heatmap showing screen results of small molecules targeting post-translational modifications in MDA-MB-468 cells. B7-H4-GFP expression was quantified by flow cytometry. NT no target. Immunoblot for B7-H4 in MDA-MB-468 (**b**), T-47D (**c**), and 4H11 (**d**) cells treated with 2-Bromopalmitic acid (2-BP) or palmostain B at the indicated concentrations for 24 h. Immunoblot for B7-H4 in mouse mammary tumor virus-polyoma middle tumor antigen (MMTV-PyMT) cells treated with 2-BP (**e**) or cerulenin (**f**) at the indicated concentration for 24 h. Biotin-acyl-exchange (ABE) detects endogenous B7-H4 palmitoylation in MDA-MB-468 (**g**) and 4H11 (**h**) cells with or without (w/o) 2-BP (100 μM) treatment for 24 h. B7-H4 palmitoylation levels were measured relative to the corresponding input levels of B7-H4 protein. Representative immunoblot for B7-H4 and protein level quantification from three independent tests in MDA-MB-468 (**i**) and 4H11 (**j**) cells examined by cycloheximide (CHX) chase assay combined w/o 2-BP (100 μM) treatment. Data are shown as mean ± SEM. Statistical analyses were performed using two-way ANOVA with Sidak's multiple comparisons test. Immunoblot for B7-H4 in MDA-MB-468 (**k**), T-47D (**l**), and 4H11 (**m**) cells treated with 2-BP and/or bafilomycin A1 (100 nM) and E64d (20 μM) plus pepstatin A (pep A) (10 μM) for 24 h. Source data are provided as Source Data File.

differentially expressed and localized across different cells[28]. To identify the predominant PAT for B7-H4, we first measured the mRNA levels of all PATs in four B7-H4[+] human breast cancer cell lines, MDA-MB-468, T-47D, SK-BR-3, and MCF7. The expression patterns of PATs exhibited similarity across different breast cell lines. Several PATs, such as ZDHHC1, ZDHHC2, ZDHHC19, and ZDHHC24, were undetectable, while others demonstrated moderate to high levels of expression in 4 cell lines (Fig. 3d). We next performed the ABE screening to compare B7-H4 palmitoylation levels among PATs in 293T cells. We found that ZDHHC3 and ZDHHC23 led to relatively high levels of B7-H4 palmitoylation (Fig. 3e and Supplementary Fig. 3c). Co-immunoprecipitation (Co-IP) analysis showed that B7-H4 physically interacted with ZDHHC3 (Fig. 3f and Supplementary Fig. 3d), but not with ZDHHC23 (Supplementary Fig. 3e). Hence, ZDHHC3 is the palmitoylation enzyme of B7-H4. As expected, overexpressing ZDHHC3 increased B7-H4 protein levels in T-47D cells, whereas knocking down ZDHHC3 decreased B7-H4 protein levels (Fig. 3g, h and Supplementary Fig. 3f). Collectively, our data indicate that ZDHHC3 is the predominant PAT for human B7-H4, mediating its palmitoylation on C130 residue.

We next investigated the role of ZDHHC3 in tumor immunity in a syngeneic immune-competent mouse model. As ZDHHC3 is responsible for adding palmitoyl moiety to B7-H4, thereby stabilizing B7-H4 protein, we theorized that the absence of ZDHHC3 could slow down tumor progression. In support of this possibility, knocking down ZDHHC3 (sh*Zdhhc3*) in 4H11 cells failed to alter tumor cell proliferation in vitro (Supplementary Fig. 3g), but reduced B7-H4 expression, and inhibited tumor growth in immune-competent mice (Fig. 3i–k and Supplementary Fig. 3h). As a result, there was an increase in both the number and percentage of tumor-infiltrating CD8[+] T cells, as well as elevated levels of IFN-γ[+] and TNF-α[+] CD8[+] T cells in sh*Zdhhc3* tumors compared to control shRNAs (Fig. 3l–n and Supplementary Fig. 3i). Knocking down ZDHHC3 also led to a decrease in PD-1[+]TIM-3[+] and TCF-1[−]TOX[+] exhausted CD8[+] T cells, and an increase in TCF-1[+]TOX[−]CD8[+] stem-like T cells (Fig. 3o–q and Supplementary Fig. 3j). These results suggest that ZDHHC3 stabilizes B7-H4 and mediates T cell suppression, thereby supporting tumor progression.

## Abemaciclib promotes B7-H4 lysosomal degradation

B7-H4 is highly expressed in breast cancer and degraded in the lysosome. Functional neutralizing B7-H4 mAb remains under development. We hypothesized that targeting lysosomal B7-H4 degradation is a rational alternative to diminish B7-H4-mediated immune suppression. Abemaciclib is an FDA-approved CDK4/6 inhibitor for breast cancer treatment. Interestingly, abemaciclib can enhance lysosome biogenesis and biomass[29,30]. In line with this, we found that abemaciclib-treated MDA-MB-468 cells exhibited an increased abundance of lysosomes as measured by LysoTracker staining in a dose-dependent manner (Fig. 4a,b). We conducted an RNA-sequencing study in MDA-MB-468 cells treated with abemaciclib. RNA-seq analysis revealed that treatment with abemaciclib induced a group of gene expression related to lysosome biogenesis and function, including TFEB (transcription factor EB, a master regulator for lysosome biogenesis), the endo-lysosome trafficking genes (including SORT1 and AP-1/2/3/4 complex), the lysosomal membrane proteins (including LAMPs, ATP6s, CD63, and MCOLN1), and

the lysosomal digestive enzymes (including CTSB, CTSK, DNASE2, and NAGLU) (Fig. 4c). The data suggests that abemaciclib primes lysosome activation and function, thereby promoting B7-H4 lysosomal degradation. To test this possibility, we treated MDA-MB-468 and T-47D cells with abemaciclib. Immunoblot experiments showed that treatment with abemaciclib reduced B7-H4 expression in a dose-dependent manner (Fig. 4d, e). A similar effect was observed in mouse cell lines Py230 and MMTV-PyMT (Supplementary Fig. 4a, b). Cell surface expression of B7-H4 is important for T cell suppression[3–5,9]. Immunoblot analysis demonstrated that cell surface B7-H4 was reduced in abemaciclib-treated cells (Fig. 4f). We further conducted the CHX chase assay in MDA-MB-468 (Fig. 4g) and Py230 cells (Supplementary Fig. 4c). Treatment with abemaciclib and CHX accelerated B7-H4 degradation as compared to CHX alone (Fig. 4g and Supplementary Fig. 4c). Abemaciclib treatment failed to change B7-H4 transcripts (Supplementary Fig. 4d). Thus, abemaciclib downregulates the B7-H4 protein. To test whether abemaciclib reduced B7-H4 protein levels via lysosomal degradation, we examined the role of abemaciclib in B7-H4 in the presence of bafilomycin A1 or the E64d plus pepstatin A. Indeed, lysosome inhibition reversed the inhibitory effect of abemaciclib on B7-H4 protein levels in MDA-MB-468 and T-47D cells (Fig. 4h, i), while MG132 failed to do so (Supplementary Fig. 4e, f). Thus, abemaciclib enhances B7-H4 lysosomal degradation. Given that the effect of abemaciclib on B7-H4 protein may occur following endocytosis prior to B7-H4 lysosomal degradation, we examined this possibility using the endocytosis assay. Flow cytometry analysis showed that abemaciclib had no obvious effect on the uptake of dextran and the surface expression of transferrin receptor (TfR) in MDA-MB-468 cells (Supplementary Fig. 4g, h). The data suggest that abemaciclib does not broadly alter cellular endocytosis.

Abemaciclib, palbociclib, and ribociclib are the widely used FDA-approved CDK4/6 inhibitors[31,32]. We wondered whether the 3 CDK4/6 inhibitors comparably impact B7-H4 protein. Unexpectedly, differed from abemaciclib, neither palbociclib nor ribociclib reduced B7-H4 protein levels in MDA-MB-468 and Py230 cells (Fig. 4j, k and Supplementary Fig. 4i, j). We confirmed this observation in additional human breast and ovarian cell lines, including SK-BR-3, T-47D, and OVCAR3 (Supplementary Fig. 4k–m). Interestingly, abemaciclib remained effective in reducing B7-H4 protein in the presence of palbociclib or ribociclib (Fig. 4j, k and Supplementary Fig. 4i, j). This suggests that inhibition of CDK4/6 may be irrelevant to abemaciclib-mediated B7-H4 reduction. This possibility is in line with the fact that MDA-MB-468 lacks expression of the Rb gene[33,34]. To further test this possibility, we manipulated the cell cycle in both MDA-MB-468 and Py230 cells using nocodazole, which targets β-tubulin and arrests the cell cycle at metaphase[35,36]. Abemaciclib effectively reduced B7-H4 expression independent of the presence of nocodazole (Fig. 4l and Supplementary Fig. 4n). Thus, abemaciclib mediates B7-H4 lysosomal degradation independent of its CDK4/6 inhibitory role.

## Abemaciclib enhances B7-H4 lysosomal degradation promoting tumor immunity

Abemaciclib targets CDK4/6 to mediate anti-tumor activities in clinical and preclinical settings[32,37]. It remains unclear whether abemaciclib can abolish tumor B7-H4 expression to enhance anti-tumor immunity in vivo. To test this, we isolated primary tumor cells from a MMTV-

 

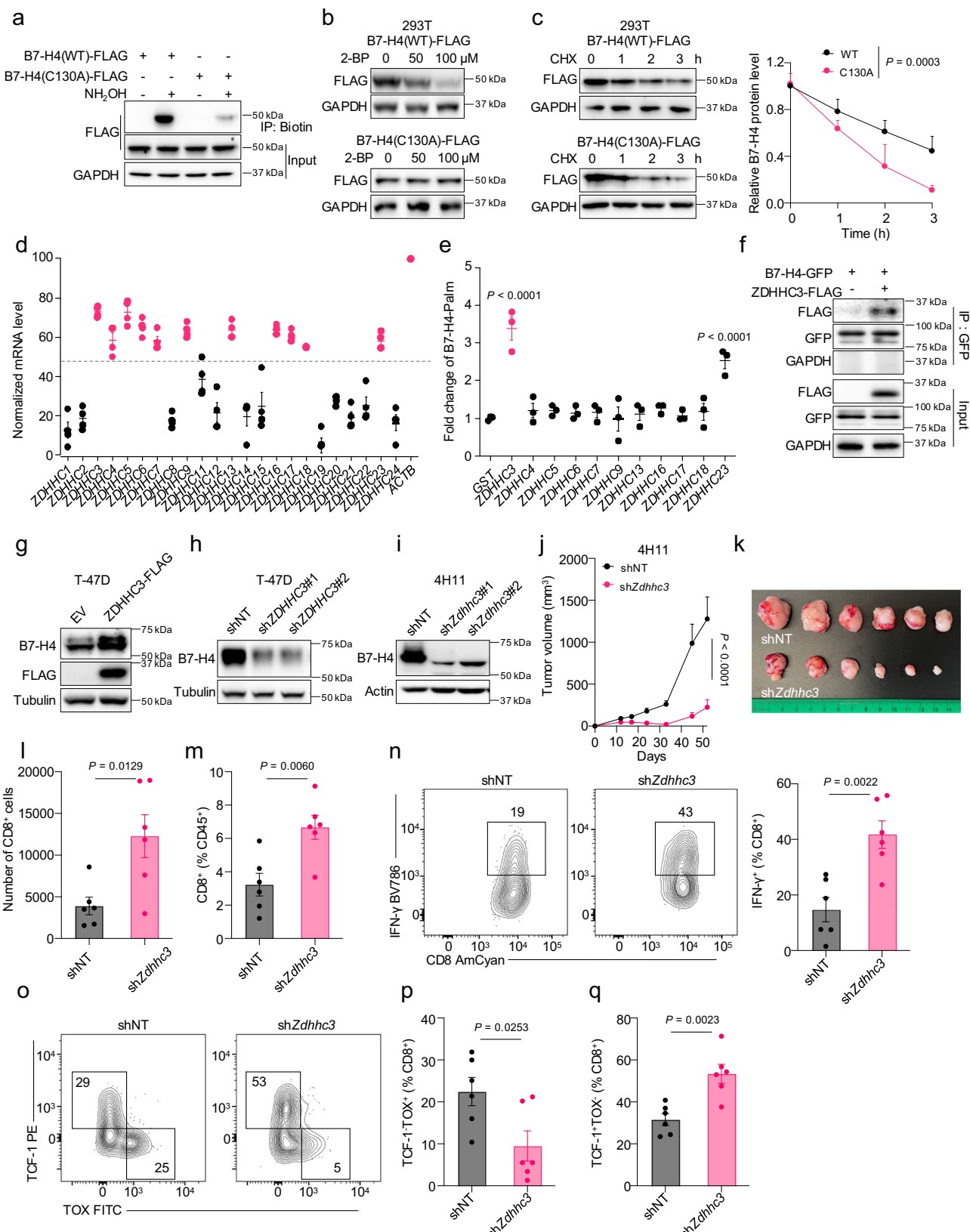

PyMT spontaneous breast cancer model[38] and established MMTV-PyMT tumor cells in syngeneic wild-type mice. Then, we treated MMTV-PyMT tumor-bearing mice with abemaciclib. Abemaciclib treatment inhibited tumor progression as shown by tumor volume (Fig. 5a, b) and tumor weight (Fig. 5c). In line with this, B7-H4 protein expression was reduced in tumor tissues (Fig. 5d, e). Flow cytometric

analysis revealed an increase in tumor-infiltrating CD8+ T cells and IFN-γ+CD8+ T cells (Fig. 5f–h). Thus, abemaciclib treatment downregulates B7-H4 in vivo and enhances the anti-tumor immune response. To broaden our work to another cancer type, we introduced B7-H4 expression in CT26 cells (B7-H4-OE) (Supplementary Fig. 5a), a mouse colorectal carcinoma cell line without endogenous B7-H4 expression.

**Fig. 3 | ZDHHC3 catalyzes B7-H4 palmitoylation at Cys130 impairing tumor immunity. a** Comparison of B7-H4-C130A palmitoylation with B7-H4-WT in 293T cells detected by ABE assay. **b** Immunoblot for B7-H4 WT and C130A in 293T cells treated with 2-BP for 24 h. **c** Representative immunoblot for B7-H4 WT and C130A and protein level quantification from three independent tests in 293T cells examined by cycloheximide (CHX) chase assay. Data are shown as mean ± SEM. Statistical analysis was performed using two-way ANOVA with Sidak's multiple comparisons test. **d** Normalized 23 human palmitoyltransferases (PATs) mRNA levels in MDA-MB-468, T-47D, SK-BR-3, and MCF7 cells using ACTB as reference. n = 4 per group. Data are shown as mean ± SEM. **e** Quantitative analysis of relative B7-H4 palmitoylation levels from three independent ABE screens by co-overexpressing the candidates PATs from (**d**) and B7-H4 in 293T cells. Data are shown as mean ± SEM. Statistical analysis was performed using one-way ANOVA with Dunnett's multiple comparisons test to GST control. **f** Interaction between B7-H4-GFP and ZDHHC3-Flag in 293T cells. Immunoblot for B7-H4 in T-47D cells with

ZDHHC3 overexpression (**g**) or knockdown (**h**) and in 4H11 cells with ZDHHC3 knockdown (**i**). shNT non-targeting shRNA. Zdhhc3 knockdown (sh*Zdhhc3*) and control (shNT non-targeting shRNA) 4H11 subcutaneous tumor growth (**j**), representative image (**k**). n = 6 per group. Subcutaneous implantation was conducted in 8-week-old female C57BL/6 mice. Data are shown as mean ± SEM (**j**). Statistical analysis was performed using two-way ANOVA (**j**). Flow cytometry analysis of the number (**l**) and percentage (**m**) of tumor-infiltrating CD8⁺ T cells, IFN-γ⁺CD8⁺ T cells (**n**) in subcutaneous 4H11 tumor-bearing mice. n = 6 per group. Subcutaneous implantation was conducted in 8-week-old female C57BL/6 mice. Data are shown as mean ± SEM. Statistical analyses were performed using two-tailed Student's t test. Flow cytometry analysis of tumor-infiltrating TCF-1⁻TOX⁺ (**o**, **p**), and TCF-1⁺TOX⁻ (**o**, **q**) CD8⁺ T cells in subcutaneous 4H11 tumor-bearing mice. n = 6 per group. Subcutaneous implantation was conducted in 8-week-old female C57BL/6 mice. Data are shown as mean ± SEM (**p**, **q**). Statistical analyses were performed using two-tailed Student's t test (**p**, **q**). Source data are provided as Source Data File.

---

Treatment with abemaciclib downregulated B7-H4 protein levels in B7-H4-OE CT26 cells in vitro (Supplementary Fig. 5b) and in tumor tissues in vivo (Supplementary Fig. 5c, d). In mice bearing CT26 control (CT26-EV) tumors, treatment with abemaciclib inhibited tumor progression (Supplementary Fig. 5e). As expected, B7-H4 overexpression accelerated tumor growth, consistent with its immune inhibitory role (Supplementary Fig. 5f–i). Notably, abemaciclib treatment resulted in greater tumor growth suppression in CT26-B7-H4-OE tumors compared to CT26-EV tumors (Supplementary Fig. 5f–i). These findings suggest that abemaciclib effectively mitigates the protumor effect of B7-H4. Altogether, apart from its CDK4/6-dependent activity, abemaciclib can lower B7-H4 protein levels, thereby boosting tumor immunity and preventing tumor progression.

## Discussion

B7-H4, a member of the B7 co-inhibitory family of ligands that antagonizes T cell functions[3–5,18]. PTMs affect the property of a protein, enabling functional adaptation of a given protein to both cell-intrinsic and extrinsic stimuli[39]. We have assessed the potential PTMs of B7-H4 and their relevance in the context of tumor immune responses. A recent report shows that B7-H4 in breast cancer cells experiences glycosylation[7]. Interestingly, we discover that palmitoylation is a distinct PTM in multiple mouse and human breast cancer cells, ensuring B7-H4 protein stability. Palmitoylation often functions as a sorting signal, directing proteins to their terminal destinations[27,40–42]. We demonstrate that palmitoylation deficiency reduces the total and cell membrane B7-H4 in breast cancer cells. It suggests that palmitoylation is required to sustain the cell membrane pool of B7-H4, ensuring its immunosuppressive function. Since B7-H4 protein instability caused by palmitoylation deficiency can be restored by sabotaging the lysosome function or its hydrolyzing enzymes, we suggest that lysosome is the degradative terminal of palmitoylation-regulated B7-H4.

We identify ZDHHC3 as the palmitoyltransferase for B7-H4, catalyzing its palmitoylation dominantly at Cys130. ZDHHC3 is primarily localized to the membrane of the Golgi apparatus[28]. Hence, B7-H4 may be palmitoylated at the Golgi apparatus. Further investigation is needed to underscore the structural and molecular mechanisms of the interaction between B7-H4 and ZDHHC3. Given that ZDHHC3 can catalyze B7-H4 palmitoylation, our results suggest that their biological effects can be partially overlapping but not identical. Additionally, it has been reported that ZDHHC3 can increase tumor cell resistance to oxidative stress and enhance tumor PD-L1 expression, contributing to tumor progression[43,44]. B7-H4 is predominantly expressed in breast cancer cells in the TME[45]. The pathological significance of the interplay between B7-H4 and ZDHHC3 in non-tumor cells in the breast cancer microenvironment remains to be studied.

Small-molecule CDK4/6 inhibitors, such as palbociclib, ribociclib, and abemaciclib, are used to treat patients with hormone receptor-positive breast cancers[29,46]. Interest has grown in understanding the

role of CDK4/6 inhibitors beyond their impact on tumor cell proliferation[47–50]. For instance, CDK4/6 inhibition can upregulate type-III IFN production and suppress regulatory T cell proliferation[49]. We demonstrate that B7-H4 is degraded in the lysosome. Previous report reveals an active role of abemaciclib in lysosome[29]. We hypothesize that abemaciclib potentially promotes B7-H4 lysosomal degradation, thereby improving anti-tumor immunity. In support of this hypothesis, we discover that treatment with abemaciclib leads to reduced tumor B7-H4 expression, enhanced T cell activation, and abolished tumor progression in mice bearing breast cancer and colorectal cancer. This anti-tumor effect is accompanied with several intriguing features. (a) Abemaciclib enhances lysosomal biomass and catabolic activity in breast cancer cells. (b) Abemaciclib diminishes total and cell membrane B7-H4 protein in breast cancer cells. (c) Abemaciclib promotes B7-H4 degradation independent of CDK4/6 and cell cycle status. (d) Although all three inhibitors target CDK4/6, leading to G1 cell cycle arrest through RB phosphorylation suppression[47], abemaciclib, but not palbociclib and ribociclib, induces B7-H4 protein degradation. These features suggest that abemaciclib acts beyond its defined inhibitory role in CDK4/6, is a unique immune modulator of B7-H4, and can be repurposed to treat patients with B7-H4⁺ tumors.

B7-H4 was identified more than 20 years ago[3–5]. Targeting B7-H4 is thought to be a promising immunotherapeutic approach for cancer therapy[18,51–53]. However, there is no FDA-approved small-molecule or antibody specifically blocking B7-H4-mediated immune inhibitory activity. FPA150, an anti-B7-H4 antibody, was tested in a phase 1 clinical trial in 2018 (NCT03514121), but the clinical results have not yet been publicly disclosed. Recently, several B7-H4-targeted antibody-drug conjugates (ADCs) have entered phase 1 clinical trials (XMG-1660: NCT05377996, AZD8205: NCT05123482, HS-20089: NCT05263479, SGN-B7H4V: NCT05194072). The principle of these trials is to use B7-H4 as a delivery system rather than directly block B7-H4-mediated immunosuppression. Given that ZDHHC3 mediates B7-H4 palmitoylation, palmitoylation stabilizes total and cell membrane B7-H4 expression, and B7-H4 is degraded in lysosome, we suggest two alternatives to target B7-H4 for cancer immunotherapy. The first is to target tumor ZDHHC3 to abolish B7-H4 palmitoylation, thereby destabilizing the B7-H4 protein. The second option is to target lysosome, leading to accelerated B7-H4 degradation. In this regard, clinically repurposing the utilization of abemaciclib may be a ready approach to treating patients with B7-H4 expressing tumors. Based on this, identifying and validating biomarkers linked to B7-H4 expression and lysosomal activity could be crucial for determining disease indications and stratifying patients for abemaciclib treatment.

## Methods

Studies in this research comply with all relevant ethical regulations required by the Institutional Ethics Committee and the Institutional Animal Care and Use Committee at the University of Michigan.

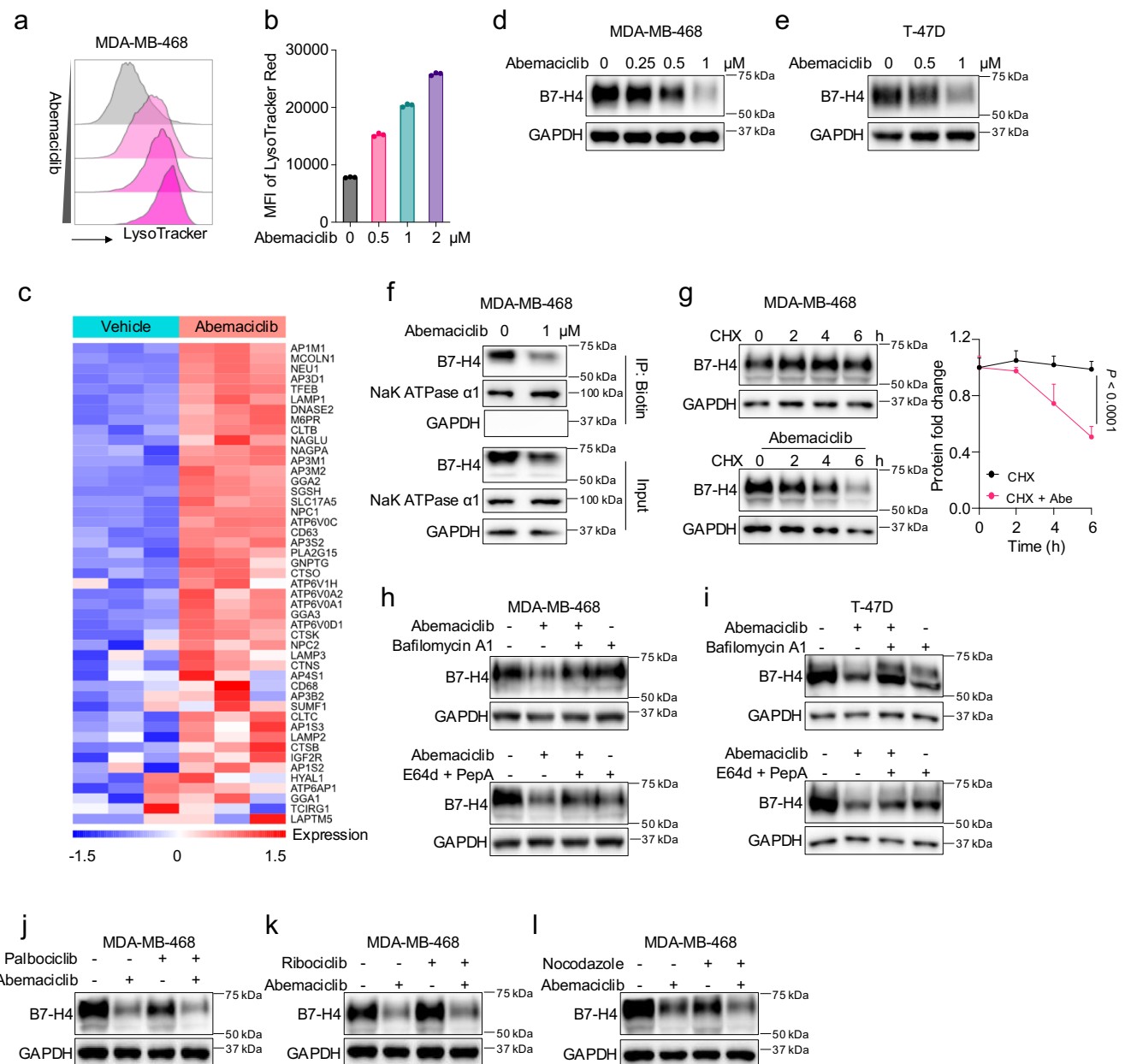

**Fig. 4 | Abemaciclib promotes B7-H4 lysosomal degradation. a, b** Representative flow cytometry analysis of LysoTrack-DND-99 staining in MDA-MB-468 cells treated with abemaciclib at the indicated concentrations (**a**). The mean fluorescent intensity (MFI) of LysoTrack-DND-99 was quantified from three parallel treatments (**b**). Data are shown as mean ± SD (**b**). **c** Normalized heatmap of MDA-MB-468 RNA sequencing following treatment with abemaciclib for 12 h. Immunoblot for B7-H4 in MDA-MB-468 (**d**) and T-47D (**e**) breast cancer cells treated with abemaciclib at the indicated concentrations for 24 h. **f** Immunoblot for cell surface B7-H4 in MDA-MB-468 cells treated with abemaciclib for 24 h. **g** Representative immunoblot for B7-H4 protein level quantification from three independent tests in MDA-MB-468 cells

examined by cycloheximide (CHX) chase assay combined w/o abemaciclib (1 μM) treatment. Data are shown as mean ± SEM. Statistical analysis was performed using two-way ANOVA with Sidak's multiple comparisons test. Immunoblot for B7-H4 in MDA-MB-468 (**h**) and T-47D (**i**) cells treated with abemaciclib (1 μM) and/or bafilomycin A1 (100 nM) and E64d (20 μM) plus pepstatin A (pep A) (10 μM) for 24 h. Immunoblot for B7-H4 in MDA-MB-468 cells treated with abemaciclib (1 μM) and/or palbociclib (1 μM) (**j**) and ribociclib (1 μM) (**k**) for 24 h. **l** Immunoblot for B7-H4 in MDA-MB-468 cells treated with abemaciclib (1 μM) and/or nocodazole (1 μM) for 24 h. Source data are provided as Source Data File.

## Ovarian cancer patient tissue samples

Individuals diagnosed with high-grade serous ovarian cancer were recruited for this study. Research materials were collected without consideration of race or ethnic origin. The study received approval from the Institutional Ethics Committee of the University of Michigan, under IRB #HUM00195340. Frozen ovarian cancer tissues were lysed and subjected to the acyl-biotin-exchange assay to detect B7-H4 expression and palmitoylation.

## Animal models and tumor establishment

Animal procedures were approved by the Institutional Animal Care and Use Committee at the University of Michigan (PRO00011876). The study involved the following mouse strains: NOD.SCID c-deficient (NSG) mice, wild-type C57BL/6 mice, wild-type BALB/c mice, and FVB/NJ (aged 6–10 weeks) (The Jackson Laboratory). All mice were housed under pathogen-free conditions.

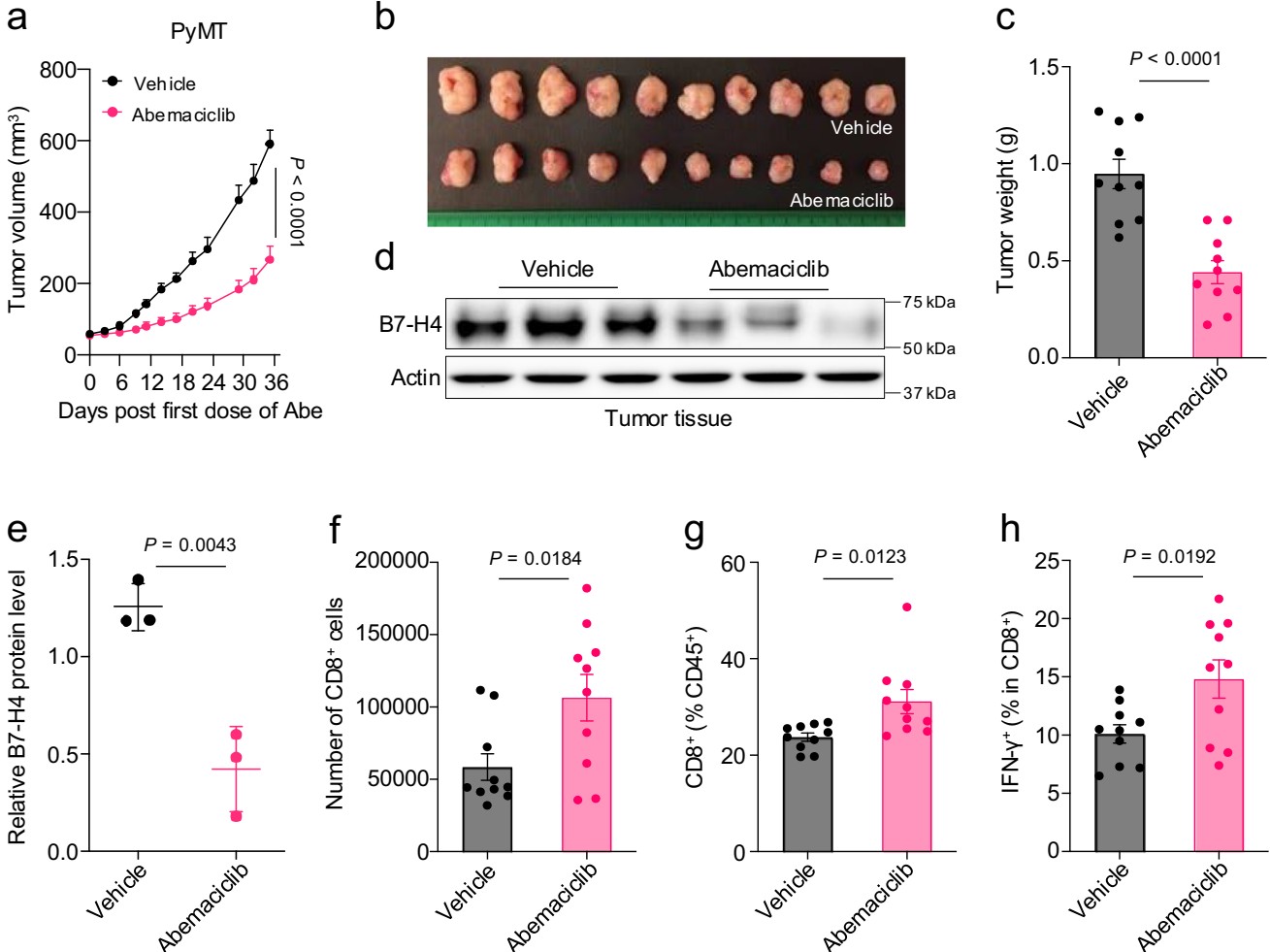

**Fig. 5 | Abemaciclib enhances B7-H4 lysosomal degradation promoting tumor immunity.** MMTV-PyMT subcutaneous tumor growth (**a**), representative image (**b**), and tumor weights (**c**) in mice treated with vehicle or abemaciclib (75 mg/kg, every 2 days). n = 10 per group. Subcutaneous implantation was conducted in 8-week-old female FVB/NJ mice. Data are shown as mean ± SEM (**a**, **c**). Statistical analyses were performed using two-way ANOVA (**a**) and two-tailed Student's *t* test (**c**). Representative immunoblot (**d**) and quantification (**e**) for B7-H4 in primary tumor tissues isolated from PyMT tumor-bearing mice treated with vehicle or abemaciclib. n = 3 per group. Subcutaneous implantation was conducted in 8-week-

old female FVB/NJ mice. Data are shown as mean ± SEM (**e**). Statistical analysis was performed using two-tailed Student's *t* test (**e**). Flow cytometry analysis of the number (**f**) and percentage (**g**) of tumor-infiltrating CD8[+], IFN-γ[+]CD8[+] (**h**) T cells in subcutaneous MMTV-PyMT tumor-bearing mice treated with vehicle or abemaciclib. n = 10 per group. Subcutaneous implantation was conducted in 8-week-old female FVB/NJ mice. Data are shown as mean ± SEM. Statistical analyses were performed using two-tailed Student's *t* test. Source data are provided as Source Data File.

To establish 4H11 tumor models (WT, B7-H4-KO, and Zdhhc3 knockdown), we subcutaneously injected $3 \times 10^6$ tumor cells into both flanks of female mice. Similarly, 4T1 control and B7-H4 overexpression tumors were established via subcutaneous injection in BALB/c. For CD8 T cell depletion, 7 days after tumor cell inoculation, mice were randomly assigned to control or treatment groups. They received intraperitoneal injections of anti-CD8 (10 mg/kg, BioXCell, BE0117) or isotype control (10 mg/kg, BioXCell, BE0090). Antibody treatments and tumor measurements occurred every 3 days.

In PyMT tumor models, $2 \times 10^6$ tumor cells were subcutaneously injected into both flanks of female FVB/NJ mice. Abemaciclib treatment was initiated 3 days post-tumor inoculation. Tumor-bearing mice were randomized into two experimental groups (vehicle and abemaciclib). Abemaciclib (75 mg/kg), dissolved in 20% PEG400 at 2 mg/mL, was administered via oral gavage every other day. For CT26 tumor models, $1 \times 10^5$ tumor cells were subcutaneously injected into both flanks of female mice. Abemaciclib treatment followed the same protocol. In our animal protocol approved by the Institutional Animal Care and Use Committee (PRO00011876), the standard tumor

endpoint is 2 centimeters; we affirm that the maximum tumor size and burden in our protocol were not surpassed. We measured tumor size with calipers and estimated tumor volume using the formula: volume = (length × width × width)/2. Following euthanasia, tumors were removed and weighed.

### Reagents

2-Bromopalmitic acid (#238422), Palmostatin B (#178501), MG132 (#474791), Hydroxylamine (#438227) and N-ethylmaleimide (E3876) were purchased from Sigma. Staurosporine (#81590), Tunicamycin (#11445), Bay 117082 (#10010266), PYR41 (#15226), 5-Deoxy-5-Methythioadenosine (#15593), Garcinol (#10566), Curcumin (#81025), Anacardic acid (#18422), Lonfarnib (#11746), Cerulenin (#10005647), Cycloheximide (#14126), E-64d (#13533), Pepstatin A (#9000469), Bafilomycin A1 (#11037), Palbociclib (PD0332991), Ribociclib(#17666), and Nocodazole (#13857) were purchased from Cayman Chemical. Abemaciclib (S5716 for in vitro) was purchased from Selleckchem.15-Azido-pentadecanoic Acid (PAA, C10265), HPDP-Biotin (A35390), AlamarBlue (DAL1025), LysoTracker Red DND-99 (L7528), Fluorescein Dextran

(D1845), and pHrodoRed Transferrin Conjugates (P35376) was purchased from Thermo Fisher Scientific. Abemaciclib (Verzenio for in vivo). IgG (BE0090) and anti-mouse CD8 (BE0117) were purchased from BioXCell.

## Cell culture

Human breast cancer cell lines MDA-MB-468 (HTB-132), SK-BR-3 (HTB-30), and T-47D (HTB-133), human ovarian cancer cell line OVCAR3 (HTB-161), mouse breast cancer cell line 4T1 (CRL-2539), MCF7 (HTB-22), and Py230 (CRL-3279), mouse colon cancer cell line CT26 (CRL-2638), and human embryonic kidney cell line HEK293T (CRL-3216) were purchased from the American Type Culture Collection (ATCC). The 4H11 cells were isolated and established from spontaneous breast tumors induced by MPA plus DMBA in C57BL/6 mice. The PyMT primary tumor cells were isolated from the mouse mammary tumor virus-polyoma middle tumor-antigen (MMTV-PyMT) tumors[38].

MDA-MB-468 cells were cultured in DMEM (Gibco, 11965−092) supplemented with 10% heat-inactivated fetal bovine serum (FBS, Gibco, 10082147) and a 1:100 supply of penicillin-streptomycin (Gibco, 15140122). T-47D, Py230, 4T1, CT26, primary tumor cells, and isolated mouse breast cancer clones 4H11 and PyMT were cultured in RPMI 1640 medium supplemented with 10% FBS and a 1:100 supply of penicillin-streptomycin. Regular testing for mycoplasma contamination was performed every 2 weeks for all cell lines.

## Cell proliferation and viability assay

Tumor cells were collected and seeded into 96-well plates. To assess the impact of treatment on cell growth and viability, a 10% volume of Alamar Blue (Thermo Fisher Scientific, DAL1025) was added directly into the medium. After 4−6 h of incubation following adhesion, absorbance at wavelengths of 570 nm and 600 nm was measured. The percentage reduction difference between treated and control cells was calculated using the following equation: Percent difference between treatment and control (%) = ((117,216 × A570 of treatment) − (80,586 × A600 of treatment))/((117,216 × A570 of control) − (80,586 × A600 of control)) × 100. After calculation, the viability of control cells was 100%, and all others were normalized to control and shown as relative cell viability (%).

## Generation of B7-H4 knockout cell line

B7-H4 knock-out cells were generated following a previously described protocol[54]. The guide RNA (gRNA) was designed using the CRISPR Design Tool (http://crispr.mit.edu) to minimize potential off-target effects. Two pairs of gRNA sequences (#1 F: 5′−CACCG GGCGGA-CAGTGCGTAAAATC-3′, #1 R: 5′−AAACGATTTTACGCACTGTCCGCCC-3′; #2 F: 5′−CACCGTTTGGATTCGACCCAGCGGT-3′, #2 R: 5′−AAA-CACCGCTGGGTCGAATCCAAAC-3′) were identified and cloned into the lentiCRISPR v2 vector (Addgene, 52961) to create sgRNA expressing vectors. Subsequently, the targeting vector was transfected into HEK293T cells with Lipofectamine 3000 (Thermo Fisher Scientific, L3000015) to produce a concentrated lentivirus preparation. Control was established using the lentiCRISPR v2 vector containing a GFP-targeting guide RNA (sgGFP). The lentivirus particles were then used to transduce the target cells. After a 48-h post-transfection, the culture medium was supplemented with puromycin (2 µg/mL; GoldBio, P-600-100). The cells were subsequently seeded at a clonal density into each well of 96-well plates, utilizing a limiting dilution technique. The knockout efficacy of B7-H4 was verified through immunoblot analysis, and a minimum of five independent B7-H4-knockout cell lines were selected for subsequent experimental procedures.

## Cloning and mutagenesis

A specific primer pair was utilized to facilitate the overexpression of mouse B7-H4 in tumor cells. This pair was designed to clone the mouse B7-H4 and insert it into a lentivirus plasmid backbone at the HpaI/XbaI sites. The forward primer sequence is 5′-AGGCTGTTAACATGGCTTCC

TTGGGGCAGACCA-3′, and the reverse primer sequence is 5′-tataatctctagaTCAtcttagcatcaggcaacag-3′. A human C-terminal GFP-tagged B7-H4 construct was procured from VectorBuilder (Chicago, USA). Following this, the plasmid underwent sequencing and validation to facilitate the generation of B7-H4-GFP expression clones in MDA-MB-468 cells.

For the detection of palmitoylation by the ABE assay in 293T cells, a C-terminal 3x Flag-tagged human B7-H4 was generated. The human wild type (WT) B7-H4 was cloned using a specific primer set and inserted into the HpaI/XbaI sites of the pHIV-EGFP (Addgene, #21373) lentivirus plasmid, which had a preinstalled 3x Flag at the C terminal. The forward primer sequence for this process is 5′−AGCGGCCGCTGAGTTAA-CATGGCTTCCCTGGGGCAGAT-3′, and the reverse primer sequence is 5′−TGGTCTTTGTAGTCTCTAGATTTTAGCATCAGGTAAGG-3′. Overlapping PCR with specific primers introduced cysteine to alanine point mutations. The resulting mutants of B7-H4 were inserted into pHIV-EGFP to replace WT B7-H4.

The primer sequences used for the point mutations are as follows:
C130AR: 5′−AGCTTTGTAGGTGCCAGCATC-3′,
C130AF: 5′−GGCACCTACAAAGCTTATATCATCACTTCTAAAGG-3′;

Mouse palmitoylacyltransferase expression plasmids were generously provided by Dr. Masaki Fukata (National Institute for Physiological Sciences, Japan). The human ZDHHC3 lentivirus expression plasmid was ordered from VectorBuilder.

## Cell surface protein isolation and detection

The isolation of biotin-labeled membrane protein was conducted as per the instructions provided by the manufacturer of the Pierce Cell Surface Biotinylation and Isolation Kit (Thermo Fisher Scientific, A44390). In brief, cells were collected and suspended in 1 × PBS buffer (Cytiva, SH30256.02) supplemented with Sulfo-NHS-SS-Biotin. This suspension was then incubated at room temperature for a duration of 10 min. Following this, the cells were centrifuged and then washed twice with 1 × TBS. A volume of 500 µL of lysis buffer was added to resuspend the pellet, followed by an incubation period on ice for 30 min. The lysate was centrifuged at 16,000 × g for 5 min, and the pellet was discarded. The supernatant was subjected to NeutrAvidin™ Agarose (Thermo Fisher Scientific, 29201) binding at room temperature for 30 min. The agarose was washed four times with the wash buffer and eluted with the elution buffer at room temperature for 30 min. Equivalent portions of different samples were subsequently subjected to immunoblot analysis.

## Immunoblotting

Cells were subjected to a wash in cold PBS and lysed using 1 × RIPA lysis buffer (Thermo Fisher Scientific, 89900) supplemented with 1 × Halt Protease Inhibitor Cocktail (Thermo Fisher Scientific, 78429). The lysates were then incubated on ice for a duration of 30 min and cleared by centrifugation at 15,000 × g for 10 min. The protein concentration was quantified using the Pierce BCA Protein Assay Kit (Thermo Fisher Scientific, 23225). Thirty milligrams of protein was mixed with sample buffer (Thermo Fisher Scientific, 1610747) and denatured at 95 °C for 10 min. The sample was then separated by SDS-PAGE and transferred to a nitrocellulose membrane (BioRad, 1620145). The membranes were blocked with 5% w/v nonfat dry milk and incubated with primary antibodies overnight at 4 °C. This was followed by incubation with HRP-conjugated secondary antibodies (Vector Laboratories, PI-9400-1, and PA-9200-1.5) for 1 h at room temperature. The signal was detected using Clarity and Clarity Max Western ECL Blotting Substrates (Bio-Rad, 1795060) and captured using the ChemiDoc Imaging System (Bio-Rad). The antibodies used were rabbit anti-B7-H4 (1:1000) (Abcam, ab209242), rabbit anti-Actin (1:1000) (Cell Signaling Technology, 4967), mouse anti-Flag (1:2000) (Genescript, A01429-100), rabbit anti-Tubulin (1:1000) (Cell Signaling Technology, 2144), mouse anti-GAPDH (1:2000) (Santa Cruz, sc-32233), rabbit anti-Vinculin

(1:1000) (Cell Signaling Technology, 4650), and rabbit anti-Na, K-APTase α1 (1:1000) (Cell Signaling Technology, 3010). All immunoblots displayed are representative of three independent experiments.

## LC-tandem mass spectrometry

Three technical replicate samples of FLAG-tagged B7-H4, alongside FLAG-only control from MDA-MB-468 cells, underwent immunoprecipitation and affinity purification. The purity of B7-H4-FLAG was assessed through electrophoresis, and the resulting samples were then stored at −80 °C. A subsequent round of collection was conducted, and these samples were submitted to the Mass Spectrometry Core at the University of Michigan Medical School's Department of Pathology for LC-Tandem MS analysis. For in-solution digestion of B7-H4-FLAG protein, beads were resuspended in 50 mL of 0.1 M ammonium bicarbonate buffer (pH = 8). Cysteines were reduced by the addition of 50 mL of 10 mM DTT and incubated at 45 °C for 30 min. Following this, samples were cooled to room temperature, and cysteine alkylation was performed by incubating with 65 mM 2-chloroacetamide in the dark for 30 min at room temperature. An overnight digestion lasting ~16 h with 1 μg of sequencing-grade, modified trypsin was conducted at 37 °C with constant shaking in a Thermomixer. The digestion was halted by acidification, and peptides were desalted using SepPak C18 cartridges according to the manufacturer's instructions (Waters). Samples were thoroughly dried using a vacufuge. The resulting peptides were dissolved in 8 mL of a solution containing 0.1% formic acid and 2% acetonitrile, from which 2 mL was resolved on a nano-capillary reverse phase column (Acclaim PepMap C18, 2 μm, 50 cm, Thermo Fisher Scientific). A gradient was applied at a flow rate of 300 nl/min over 90 min, using 0.1% formic acid/2% acetonitrile (Buffer A) and 0.1% formic acid/95% acetonitrile (Buffer B) (2–22% buffer B over 65 min, 22-40% over 15 min, and 40–90% over 5 min, followed by holding at 90% buffer B for 5 min and re-equilibration with Buffer A for 25 min). The eluent was directly introduced into an Orbitrap Fusion tribrid mass spectrometer (Thermo Fisher Scientific, San Jose, CA) via an EasySpray source. MS1 scans were performed at a resolution of 120 K (AGC target = $2 \times 10^5$; max IT = 100 ms). Data-dependent high-energy C-trap dissociation MS/MS spectra were collected using the top speed method (3 s) after each MS1 scan (NCE-32%; AGC target $5 \times 10^4$; max IT 50 ms, 15K resolution). Proteins were identified by comparing the MS/MS data to the H. sapiens database (UniProt) using Proteome Discoverer (v2.4, Thermo Scientific). The search parameters included an MS1 mass tolerance of 10 ppm and a fragment tolerance of 0.2 Da; two missed cleavages were permitted; carbamidomethylation of cysteine was the fixed modification, while oxidation of methionine and deamidation of asparagine and glutamine and palmitoylation of cysteine were treated as potential modifications. The false discovery rate (FDR) was calculated using Percolator, and proteins/peptides with an FDR of ≤1% were selected for further analysis.

## Immunoprecipitation

Cells were washed twice with ice-cold 1 × PBS (Cytiva, SH30256.02) and lysed in Pierce IP Lysis Buffer (Thermo Fisher Scientific, 87787) with 1 × Halt Protease and Phosphatase Inhibitor Cocktail (Thermo Fisher Scientific, 78440) on ice for 30 min. The supernatant was collected after centrifugation at $15,000 \times g$ for 15 min at 4 °C. Five hundred micrograms of cell lysate was precleared using 20 μL of Protein A/G Plus-Agarose (Santa Cruz, sc-2003). This was then incubated with GFP (Proteintech, #gtma) or Pierce Anti-DYKDDDK Magnetic Agarose Beads (Thermo Fisher Scientific, A36797) directly for a period of 2 h. After three washes with 1 mL of IP buffer and one wash with PBST, the proteins bound were released by boiling in 30 μL of SDS loading buffer. These were then detected by immunoblot.

## Click-iT Assay

Cells were treated with 100 μM 15-Azidopentadecanoic Acid (Thermo Fisher Scientific, C10265) and incubated at 37 °C for 6 h. Following

incubation, the medium was discarded, and the cells were rinsed with PBS twice. A lysis buffer (containing 1% sodium dodecyl sulfate in 50 mmol/L Tris-HCl, pH 8.0, along with protease and phosphatase inhibitors) was then added. The cell lysates were subjected to a 30-min incubation on ice, followed by sonication using a probe sonicator, vortexing for 1 min, and centrifugation at $18,000 \times g$ at 4 °C for 5 min. The supernatants were subsequently transferred to a separate tube, and the protein concentration was determined using the Pierce BCA Protein Assay Kit (Thermo Fisher Scientific, 23225). The protein samples underwent a reaction with biotin using the Click-iT Protein Reaction Buffer Kit (Thermo Fisher Scientific, C10276). The resulting biotin alkyne-azide-palmitic acid-protein complexes were isolated using streptavidin beads (Thermo Fisher Scientific, 20347). The beads were then subjected to immunoblotting to detect B7-H4 or FLAG.

## Acyl-biotin-exchange Assay

The palmitoylation of B7-H4 was evaluated using the previously described acyl-biotin exchange (ABE) assay[55]. Cells with the indicated treatment were lysed in a buffer (containing 50 mM HEPES pH 7.4, 150 mM NaCl, 5 mM EDTA, 1% Triton X-100 (v/v), and 1 × Halt Protease and Phosphatase Inhibitor Cocktail (Thermo Fisher Scientific, 78440)) at 4 °C for 30 min. Solubilized protein was precipitated using chloroform-methanol (1 : 3 : 2 protein : methanol : chloroform; CM), and the protein pellets were resuspended in a buffer containing 4% SDS (4SB; 4% SDS, 50 mM HEPES pH 7.4, 5 mM EDTA). This was diluted five-fold with a lysis buffer containing 25 mM N-Ethylmaleimide (NEM, Sigma, E3876) and incubated at 4 °C overnight with gentle agitation. Excess NEM was removed by three sequential CM precipitations, followed by resuspension in 300 μL 4SB. Samples were divided into two equal portions. One portion was diluted five-fold in a buffer (containing 0.2% Triton X-100, 1 × Halt Protease and Phosphatase Inhibitor Cocktail) with 1 M hydroxylamine (HAM, Thermo Fisher Scientific, 438227) and 1 mM EZ-link HPDP-biotin (Thermo Fisher Scientific, A35390) to cleave thioester bonds and allow the incorporation of a biotin moiety at exposed sulfur atoms. The other portion was diluted five-fold in the same buffer, replacing HAM with Tris (50 mM pH 7.4) as a control. In the absence of HAM, palmitate groups are not removed, thereby preventing biotinylation-mediated purification. Each portion was incubated at room temperature for 1 h with gentle agitation. Unreacted biotin was removed by three sequential CM precipitation, and the protein pellet was resuspended in 4SB and diluted ten-fold in Tris-containing lysis buffer. Biotinylated proteins were affinity purified using streptavidin beads by incubation at room temperature for 1 h. Bound proteins were eluted from washed beads with SDS sample buffer and subjected to immunoblot analysis.

## Flow cytometry analysis (FACS)

Single-cell suspensions were derived from fresh mouse tumor tissues. Briefly, tumor tissues were cut into small pieces and physically passed through 100 μm strainers. Immune cells were enriched by density gradient centrifugation with Ficoll (StemCell, 07851). Cells were collected, washed, and then stained with fluorescently conjugated antibodies. For cytokine detection, cell suspensions were incubated in culture medium containing PMA (5 ng/ml), ionomycin (500 ng/ml), Brefeldin A (1:1000), and Monensin (1:1000) at 37 °C for 4 h. These cells were subjected to cell surface and intracellular staining. The following antibodies were used: mouse CD45 (Clone 30-F11, BD Biosciences), CD90 (Clone 53-2.1, BD Biosciences), CD3 (Thermo Fisher Scientific, 16-0031-82), and CD8 (Clone 53-6.7, BD Biosciences), mouse IFN-γ (Clone XMG1.2, BD Biosciences), TNF-α (Clone MP6-XT22, BD Biosciences), TCF-1 (Clone S33-966, BD Biosciences), TOX (Clone REA473, Miltenyi). All flow samples were acquired through LSRFortessa, and the data were analyzed using BD FACS Diva software (BD Biosciences) and FlowJo.

### Endocytosis assay

Cellular endocytosis was assessed via the expression of plasma membrane transferrin receptor (TfR) and the uptake of fluorescein dextran. MDA-MB-468 cells were exposed to abemaciclib for 20 h. The mean fluorescent intensity (MFI) of PE-TfR (Thermo Fisher Scientific, P35376) was analyzed by FACS. Or, these abemaciclib-treated MDA-MB-468 cells were washed three times with prewarmed PBS and further incubated with a prewarmed medium containing fluorescein dextran (100 μg/mL) (Thermo Fisher Scientific, D1845) and abemaciclib. FITC-dextran positive cells were analyzed by FACS at intervals of 0.75, 2, and 9 h.

### Transcriptional profiling by RNA-seq analysis

Total RNA was isolated from cells by column purification (Direct-zol RNA Miniprep Kit, Zymo Research, R250) with DNase treatment. The Ribo-Zero Gold rRNA Removal Kit (Illumina) and TruSeq Stranded Total RNA Library Prep Globin kit (Illumina) were used to prepare the library for RNA sequencing. BGI Genomics performed sequencing and analysis. Cufflinks/CuffDiff (version 2.2.1) was used for expression quantification, normalization, and differential expression analysis. Locally developed scripts were used to format and annotate the differential expression data output from CuffDiff. The heatmap was generated using the CummeRbund R package.

### RT−qPCR analysis

Total RNA was extracted from the cell samples subjected to the indicated treatment using a Direct-zol RNA Miniprep Kit (Zymo Research, R2050) per the manufacturer's instructions. Subsequently, 0.5–1 μg of total RNA was reverse transcribed using the RevertAid First Strand cDNA Synthesis Kit (Thermo Fisher Scientific, K1621). The messenger RNA levels were measured with gene-specific primers using the SYBRTM Green PCR Master Mix (Invitrogen, 4309155) on a QuantStudio 3 Real-Time PCR System (Thermo Fisher Scientific). Fold changes in mRNA expression were calculated using the ΔΔCt method with ACTB as the reference control. The results are expressed as fold changes normalized to the controls. The primers used were human B7-H4, forward: 5′−GCAGATCCTCTTCTGGAGCATAA-3′, reverse: 5′−CCGACAGCTCATCTTTGCCTT-3′. Human ACTB forward: 5′−AGAGCTACGAGCTGCCTGAC-3′, reverse: 5′−AGCACTGTGTTGGCGTACAG-3′. Human ZDHHC3, forward: 5′−CCAGAGAAGTGTGTCCCACC-3′, reverse: 5′−GGCATTTCCTTTGGGCACTG-3′. Mouse Zchhc3, forward: 5′−TGGTGGGATTCCACTTCCTGCA-3′, reverse: 5′−GCCTCAAAGCACAGCAGGATGA-3′.

To evaluate the expression levels of PATs in the four human cancer cell lines we tested, we designed specific primers (Supplementary Data 1) for quantitative PCR (qPCR) analysis. Each primer pair underwent evaluation for amplification efficiency and melting curve analysis to ensure optimal performance. The relative expression of PATs was calculated using the following equation:

$$\text{Relative expression} = 100 + 5 \times \left( \text{Average} \left( \text{Average} \left( Ct_{ACTB} \right) - Ct_{PATs} \right) \right).$$

### Statistics and reproducibility

The representative Western blot results were derived from at least three independent biological experiments. The data for mouse and in vitro experiments were obtained from experiments that were repeated at least two times. The sample number (n) represents the number of mice or repetitions of cellular experiments, as detailed in the figure legends. Two-sided Student's $t$ tests were utilized to compare paired or independent samples, one-way analysis of variance (ANOVA) followed by Tukey's or two-sided ANOVA followed by Sidak's multiple-comparison tests as indicated, and an adjusted $P < 0.05$ was established as the cut-off. Details of the statistical analysis are provided in the corresponding figure legends. Prism v.10.0.1 (GraphPad Software) was employed to generate graphs and calculate statistics using

appropriate statistical tests depending on the data, including two-tailed Student' $t$ test and one-way or two-way ANOVA. Adjusted P values were evaluated using suitable correction methods, such as Tukey, Sidak, and Geisser–Greenhouse.

### Reporting summary

Further information on research design is available in the Nature Portfolio Reporting Summary linked to this article.

## Data availability

The transcriptomic data generated in this study have been deposited in the Gene Expression Omnibus (GEO) under accession number GSE272492. The comprehensive mass spectrometry results of B7-H4 post-translational modifications have been submitted to the ProteomeXchange Consortium, identified by PXD054393. The remaining data are available within the Article, Supplementary Information or Source Data file. Source data are provided with this paper.

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

## Acknowledgements

This work was supported in part by NIH/NCI R01 grants (CA217648, CA123088, CA099985, CA193136, and CA152470) and the NIH/NCI through the University of Michigan Rogel Cancer Center (CA46592) to W.Z. We thank the support of the University of Michigan Proteomics Resource Facility (PRF) for conducting the mass spectrometry experiments. We thank Dr. Stephen Weiss at the University of Michigan for providing the MMTV-PyMT tumors. We thank Dr. Daniel F. Hayes, Dr. Monica L. Burness, and Dr. Erin F. Cobain for insightful discussion. We are grateful for the support from Dr. Masaki Fukata at the National Institute for Physiological Sciences, Japan, who provided palmitoyltransferase expression plasmids for this work.

## Author contributions

Y. Y., conceptualization, formal analysis, validation, investigation, methodology, writing–original draft, writing–review and editing. J. Y., conceptualization, formal analysis, investigation, methodology, writing–original draft, writing–review and editing. W. W., software, investigation, review. Y. X., conceptualization, investigation, review. K. T., investigation, review. R. X., investigation, review. S. G., methodology. S. W., methodology. L. V., methodology. M. W., conceptualization. I. K., conceptualization, formal analysis, supervision, validation, investigation,

methodology, writing–review and editing. W. Z., conceptualization, formal analysis, supervision, funding acquisition, investigation, writing–original draft, writing–review and editing.

## Competing interests

W.Z. has served as a scientific advisor or consultant for Cstone, Next-Cure, and Hanchorbio. Other authors declare no competing interests.
