## [Transparent Peer Review file · Nature Communications]

Palmitoylation prevents B7-H4 lysosomal degradation sustaining tumor immune evasion

Corresponding Author: Dr Weiping Zou

Version 0:

Reviewer comments:

Reviewer #1

(Remarks to the Author)

In this manuscript, the authors aimed to understand the impact of post-translational modification of B7-H4 in cancer immunity. B7-H4 is a member of the B7 family known to be a T cell checkpoint that is important for T cell dysfunction in cancers. Recently the same group has reported an interesting role for B7-H4 in Onco-fetal immune tolerance (PMID: 38968937). Here this manuscript described an important role of palmitoylation at Cys130 that evades lysosomal degradation to promote immune suppression. Different cellular sources of tumor cells from both human and mouse were used. However, there are major concerns that relate to the specific targeting B7-H4 with palmitoylation or lysosome modification for cancer immunotherapy.

Major points: Though ZDHHC3 ablation can affect palmitoylation to change B7-H4 expression, the immunomodulatory effect of ZDHHC3 cannot necessarily be mediated by B7-H4, given that ZDHHC3 is important for palmitoylation of many proteins. Similarly, if abemaciclib can control lysosome activity, its effect can't be limited to B7-H4. Thus, the immunomodulatory effect for ZDHHC3 or abemaciclib can be solely via B7-H4.

Other points: How is the post-translational modification for B7-H4 protein in normal tissues that express B7-H4 protein, such as ductal and mucosal epithelial cells of the GI tissues?

Reviewer #2

(Remarks to the Author)

This study presents a search and analysis of the modification of B7-H4 for immune checkpoint.

The authors show that ZDHHC3 palmitoylates B7-H4 at Cys130 in breast cancer cells. Palmitoylated B7-H4 prevents lysosomal degradation and maintains B7-H4-mediated immune suppression. Knockdown of ZDHHC3 in tumor models shows in potent anti-tumor immunity and inhibits tumor progression in mice. In addition, the CDK4/6 inhibitor abemaciclib activates lysosomes and promotes B7-H4 degradation independent of the tumor cell cycle. This work is expertly done and generally clearly presented, and is therefore a valuable addition to the field. However, there are some concerns that need to be addressed to strengthen the conclusion.

The authors found that lysosomal activation by abemaciclib promotes lysosomal degradation of B7-H4. However, it is unlikely that an increase in lysosomal enzymes or the number of lysosomes would directly increase the degradation of a plasma membrane protein such as B7-H4. In fact, lysosomal degradation of plasma membrane proteins is not essential for their functional defect, since they are functionally inactivated once they are internalized. Therefore, the promotion of B4-H7 degradation by abemaciclib may include the promotion of endocytosis. It needs to be investigated whether abemaciclib promotes endocytosis, for example using dextran or DQ-BSA. Since endocytosis of plasma membrane proteins is often triggered by ubiquitination, is B7-H4 not ubiquitinated?

From the present results, it is clear that C130 of B7-H4 is palmitoylated. Interestingly, C130 of B7-H4 is located in the extracellular region. On the other hand, palmitoylation is generally a modification that occurs in the cytoplasmic region, and ZDHHC3 catalyzes palmitoylation in the cytoplasm. The authors need to explain how they think palmitoylation occurs in the

extracellular region. For example, is there a part of ZDHHC3 that is topologically oriented in the opposite direction?

The authors showed that the protein amount of B7-H4 increases with 24 h treatment with palmostatin B, indicating that B7-H4 is constitutively degraded. Does bafilomycin A1 treatment increase the protein amount as much as palmostatin B? Figures 2k, 2l and 2m showed B7-H4 in bafilomycin A1 treated cells, but the bands are saturated and not clear. It would be better to show the unsaturated bands and quantify the bands.

Reviewer #3

(Remarks to the Author)

The manuscript offers novel insights into the regulation of B7-H4 stability via palmitoylation and its role in tumor immune evasion, with a particular focus on ZDHHC3's involvement in this process. The study is well-designed and highlights an important, though underexplored, post-translational modification with potential therapeutic implications. The authors provide moderate evidence supporting the palmitoylation of B7-H4 at Cys130 and the role of ZDHHC3 in maintaining its stability. Additionally, the CDK4/6 inhibitor abemaciclib is discussed in relation to B7-H4 protein stability; however, it remains unclear whether this effect is dependent on palmitoylation. While the findings are compelling, several key areas require further clarification.

Major Comments:

1. In Figures 2g and 2h, the input level of B7-H4 in the 2-BP treated group is noticeably lower compared to the untreated group, which may significantly affect the accurate assessment of palmitoylated B7-H4 levels. Although 2-BP is a broad-spectrum palmitoylation inhibitor, there is insufficient evidence to conclude that it impacts B7-H4 protein stability through palmitoylation.
2. In Figure 3a, the input level of B7-H4 in the C130A mutant group is obviously lower relative to the wild-type (WT) group, which raises concerns about concluding that C130 is definitively the palmitoylation site of B7-H4. While the study identifies Cys130 as the palmitoylation site based on mass spectrometry and mutagenesis studies, how certain are we that no other cysteine residues on B7-H4 undergo palmitoylation? Furthermore, how might palmitoylation at other cysteine sites impact the stability or function of B7-H4? Lastly, what is the potential mechanism by which palmitoylation at Cysteine 130 regulates B7-H4 protein stability?
3. When using the $\Delta\Delta C_t$ method to calculate fold changes in mRNA expression, it is essential that the primers used for a specific target gene, such as ZDHHC, are designed to amplify the same region of the transcript. If different ZDHHC primers are used, it can result in amplification of different regions of the mRNA, which may not be directly comparable. Differences in primer efficiency, binding affinity, and the region of the gene targeted can lead to variability in C_t values, potentially skewing the results.
4. The study shows that abemaciclib, but not other CDK4/6 inhibitors like palbociclib or ribociclib, promotes B7-H4 lysosomal degradation. What mechanistic explanations might account for this difference, and how can the role of abemaciclib independent of CDK4/6 inhibition be further validated?
5. According to the paper, abemaciclib promotes the lysosomal degradation of B7-H4 but does not directly influence its palmitoylation. Rather, the study demonstrates that abemaciclib enhances lysosomal biogenesis and increases lysosomal capacity, leading to the degradation of B7-H4 in cancer cells. The palmitoylation of B7-H4 plays a crucial role in stabilizing the protein by protecting it from lysosomal degradation. Abemaciclib, by promoting lysosomal activity, counteracts this stabilization, ultimately reducing B7-H4 levels despite its palmitoylation.
6. The paper suggests repurposing abemaciclib to treat B7-H4+ tumors based on preclinical models. What would be the major challenges in translating this finding to clinical trials, and how might patient-specific factors, such as variations in B7-H4 expression or lysosomal activity, impact the efficacy of this approach?
7. Abemaciclib demonstrates strong efficacy in vivo by reducing tumor burden. How can the authors be convinced that this tumor growth attenuation occurs independently of CDK4/6 inhibition?

Reviewer #4

(Remarks to the Author)

B7-H4, a ligand from the B7 family with immunosuppressive functions, is identified as a potential target for cancer immunotherapy. In this study, Yan et al. found that ZDHHC3, a zinc finger DHHC-type palmitoyltransferase, catalyzes the palmitoylation of B7-H4 at Cys130. This modification prevents lysosomal degradation, thereby maintaining B7-H4 stability and enhancing its immunosuppressive effects, which promotes tumor growth. The researchers demonstrated that abemaciclib, a CDK4/6 inhibitor, could be repurposed to degrade B7-H4 by activating lysosomes, independently of its effects on the cell cycle. This action promotes T cell activation and counteracts B7-H4-mediated immune suppression. This is an elegant study that is robustly designed, well conducted, and clearly written, providing valuable insights to enhance our understanding of PTM in immuno-oncology.

This reviewer only has a few comments that require some clarification.

- (1) The authors convincingly demonstrated the antitumor immune activation resulting from targeting the palmitoylation of B7-H4 in tumor cells. Are there other cells, such as macrophages or neutrophils, within the TME that express B7-H4? Would targeting ZDHHC3 or abemaciclib reduce B7-H4 levels on those cells?
- (2) B7-H4 palmitoylation is a novel finding. It is somewhat unclear how generalizable this is. Are there any data from patient samples to support this?

Version 1:

Reviewer comments:

Reviewer #1

(Remarks to the Author)

The authors have adequately addressed my comments in the revised manuscript. I have no further concerns. Congratulations to the research team!

Reviewer #2

(Remarks to the Author)

In the revised manuscript the authors addressed all concerns requested by this reviewer and also added some additional experiments, which clarified and reinforced their observations.

Reviewer #3

(Remarks to the Author)

The revised manuscript is meticulously organized, and all concerns raised by the reviewers have been thoroughly and effectively addressed.

Reviewer #4

(Remarks to the Author)

The revised manuscript addressed my previous concerns. I have no further comments.

Reviewer #1 (Remarks to the Author)

In this manuscript, the authors aimed to understand the impact of post-translational modification of B7-H4 in cancer immunity. B7-H4 is a member of the B7 family known to be a T cell checkpoint that is important for T cell dysfunction in cancers. Recently the same group has reported an interesting role for B7-H4 in Onco-fetal immune tolerance (PMID: 38968937). Here this manuscript described an important role of palmitoylation at Cys130 that evades lysosomal degradation to promote immune suppression. Different cellular sources of tumor cells from both human and mouse were used. However, there are major concerns that relate to the specific targeting B7-H4 with palmitoylation or lysosome modification for cancer immunotherapy.

Response:

We appreciate your thorough review and valuable feedback on our manuscript. Below are our detailed responses to your specific points.

Major points: Though ZDHHC3 ablation can affect palmitoylation to change B7-H4 expression, the immunomodulatory effect of ZDHHC3 cannot necessarily be mediated by B7-H4, given that ZDHHC3 is important for palmitoylation of many proteins.

Response:

We focused our study on the role of ZDHHC3 in the palmitoylation of B7-H4 and its subsequent impact on B7-H4 expression and immune modulation. We concur that ZDHHC3, as a palmitoyl acyltransferase, is involved in the palmitoylation of multiple proteins, and its immunomodulatory effects may extend beyond its role in B7-H4. In response to your comment, we analyzed the effects of genetically reducing ZDHHC3 and B7-H4 on tumor progression using wild-type (immune-competent) mouse models (Fig. 1a and Fig. 3j). Our findings indicate that targeting either ZDHHC3 or B7-H4 leads to a reduction in tumor progression. Alongside our mechanistic results regarding the role of ZDHHC3 in B7-H4 expression, these complementary data suggest that ZDHHC3 may influence tumor progression through its regulation of B7-H4.

To directly address your point, we aimed to investigate whether the reduced tumor progression observed in ZDHHC3 knockdown tumors could be partially attributed to B7-H4. To this end, we attempted to create tumor lines with simultaneous genetic reductions of both ZDHHC3 and B7-H4, which would facilitate a direct comparison of their interdependent effects on tumor progression in vivo. Unfortunately, despite our multiple attempts, we were unable to successfully establish tumors in vivo with these tumor cell lines. We have revised the manuscript to clearly acknowledge this and discuss other potential targets of ZDHHC3 in tumor immune responses (Page 9, lines 270-274).

Similarly, if abemaciclib can control lysosome activity, its effect can't be limited to B7-H4. Thus, the immunomodulatory effect for ZDHHC3 or abemaciclib can be solely via B7-H4.

Response:

Abemaciclib is a well-defined CDK4/6 inhibitor, and we agree that its effects extend beyond B7-H4. Our findings indicate that treatment with abemaciclib reduces CT26 tumor progression in vivo, even though CT26 tumor cells do not express endogenous B7-H4 (see Figure 1a_to_reviewer and new Extended Data Fig. 5e, g, h). Notably, when we introduced B7-H4 into CT26 cells, we observed a significant increase in tumor inhibition mediated by abemaciclib (Figure 1b,c_to_reviewer, new Extended Data Fig. 5f-h). These results suggest that abemaciclib can effectively counteract the immune inhibition associated with B7-H4. We have incorporated these data into the revision (new Extended Data Fig. 5). We have clearly articulated that while abemaciclib influences tumor immunity through B7-H4, its effects are not limited to this pathway (Page 8, lines 245-252).

Figure 1_to_reviewer

Figure 1_to_reviewer. Abemaciclib suppresses CT26 tumor progression in the presence and absence of tumor B7-H4. Panels a-c show CT26 tumor growth curves in mice: (a) CT26-EV, (b) CT26-B7-H4, and (c) the combined growth curves. Mice were treated with either vehicle or abemaciclib (75 mg/kg, every two days), with $n \geq 8$ per group. Data are shown as mean \pm SEM. **** $p < 0.0001$, two-way ANOVA.

Other points: How is the post-translational modification for B7-H4 protein in normal tissues that express B7-H4 protein, such as ductal and mucosal epithelial cells of the GI tissues?

Response:

To address your point about the post-translational modification (PTM) of B7-H4 protein in normal tissues, we detected high levels of B7-H4 protein in uterus among several normal mouse tissues (and organs) (Figure 2a_to_reviewer). Interestingly, ABE assay demonstrated obvious B7-H4 palmitoylation in normal mouse uterine tissue (Figure 2b_to_reviewer). We have included this finding in the revised manuscript (new Extended Data Fig. 2j) (Page 5, lines 119-122).

Figure 2_to_reviewer

Figure 2_to_reviewer. Palmitoylation of B7-H4 in mouse uterine tissue. a, Immunoblot shows B7-H4 expression in mouse tissues. **b,** Biotin-acyl-exchange (ABE) reveals endogenous B7-H4 palmitoylation in mouse uterus.

Reviewer #2 (Remarks to the Author)

This study presents a search and analysis of the modification of B7-H4 for immune checkpoint.

The authors show that ZDHHC3 palmitoylates B7-H4 at Cys130 in breast cancer cells. Palmitoylated B7-H4 prevents lysosomal degradation and maintains B7-H4-mediated immune suppression. Knockdown of ZDHHC3 in tumor models shows in potent anti-tumor immunity and inhibits tumor progression in mice. In addition, the CDK4/6 inhibitor abemaciclib activates lysosomes and promotes B7-H4 degradation independent of the tumor cell cycle. This work is expertly done and generally clearly presented, and is therefore a valuable addition to the field. However, there are some concerns that need to be addressed to strengthen the conclusion.

Response:

We appreciate your positive and constructive comments on our work. Please see our detailed responses to your points.

The authors found that lysosomal activation by abemaciclib promotes lysosomal degradation of B7-H4. However, it is unlikely that an increase in lysosomal enzymes or the number of lysosomes would directly increase the degradation of a plasma membrane protein such as B7-H4. In fact, lysosomal degradation of plasma membrane proteins is not essential for their functional defect, since they are functionally inactivated once they are internalized. Therefore, the promotion of B7-H4 degradation by abemaciclib may include the promotion of endocytosis. It needs to be investigated whether abemaciclib promotes endocytosis, for example using dextran or DQ-BSA. Since endocytosis of plasma membrane proteins is often triggered by ubiquitination, is B7-H4 not ubiquitinated?

Response:

We appreciate your insightful comment regarding the lysosomal degradation of plasma membrane proteins, which typically necessitates prior endocytosis. We agree that the depletion of B7-H4 may occur after its internalization but before lysosomal degradation. To investigate whether abemaciclib promotes endocytosis as part of the B7-H4 degradation pathway, we conducted two endocytosis assays using fluorescently labeled dextran and the transferrin receptor (TfR) as markers. These assays showed that abemaciclib did not affect the endocytosis of dextran or TfR (Figure 3_to_reviewer).

It has been shown that ubiquitination facilitates the proteasomal degradation of B7-H4¹. We have included the methods and results of endocytosis (new Extended Fig. 4g,h) in the revised manuscript. (Page 7, lines 212-217; Page 15, lines 507-514).

Figure 3_to_reviewer

Figure 3_to_reviewer. Effect of abemaciclib on endocytosis in MDA-MB-468 cells. MDA-MB-468 cells were treated with or without abemaciclib (Abe) at the indicated concentrations for 20 hours. Flow cytometry shows the uptake of FITC-Dextran (100 $\mu\text{g}/\text{mL}$) (a) and the mean fluorescence of intensity (MFI) of membrane surface transferrin receptor (TfR) (b) in MDA-MB-468 cells at the indicated concentrations. n = 3.

From the present results, it is clear that C130 of B7-H4 is palmitoylated. Interestingly, C130 of B7-H4 is located in the extracellular region. On the other hand, palmitoylation is generally a modification that occurs in the cytoplasmic region, and ZDHHC3 catalyzes palmitoylation in the cytoplasm. The authors need to explain how they think palmitoylation occurs in the extracellular region. For example, is there a part of ZDHHC3 that is topologically oriented in the opposite direction?

Response:

You request our explanation regarding the relationship between the B7-H4 palmitoylation site (Cys130) and the localization of ZDHHC3. ZDHHC3 is primarily localized to the ER and/or the Golgi membranes². B7-H4 is a single transmembrane protein. B7-H4 is likely co-translated on the ER membrane and subsequently trafficked through the Golgi. It is plausible that B7-H4 is palmitoylated at the ER/Golgi compartments before its plasma membrane localization. Most palmitoyl acyltransferases (PATs) are predicted to have cytoplasmic DHHC domains. It is thought that palmitoylation generally occurs in the cytoplasmic region. Notably, only two PATs, hZDHHC20 and zfDHHS15, have been structurally characterized^{3,4}. The structural and functional diversity of the DHHC family members, including ZDHHC3, remains largely unexplored. Certain PATs may have catalytic domains within the lumen of the ER or Golgi apparatus, thereby supporting a potential palmitoylation within these organelles' luminal compartments.

The authors showed that the protein amount of B7-H4 increases with 24 h treatment with palmostatin B, indicating that B7-H4 is constitutively degraded. Does bafilomycin A1 treatment increase the protein amount as much as palmostatin B? Figures 2k, 2l and

2m showed B7-H4 in bafilomycin A1 treated cells, but the bands are saturated and not clear. It would be better to show the unsaturated bands and quantify the bands.

Response:

We agree that our data implies B7-H4 is consistently degraded in the lysosome. It is reasonable that lysosomal inhibitors could mimic the effect of other drugs that suppress B7-H4 degradation, thereby increasing its expression. Due to different specificity, EC50, treatment concentrations of the inhibitors used, and different tumor cell lines, it would be challenging to draw a conclusion about the comparative efficacy of palmostatin B and bafilomycin A1 in increasing B7-H4 protein levels. Nevertheless, we quantitatively compared the effects of palmostatin B and bafilomycin A1 on B7-H4 protein levels in MDA-MB-468 cells (related to Fig. 2b and 2k). We found that both compounds comparably upregulated B7-H4 protein levels in MDA-MB-468 cells (Figure 4a,b to_reviewer).

We used ImageLab software to re-analyze and quantify the Western blot band images from the original Fig. 2k-m. The B7-H4 bands were not saturated as shown in Figure 4c-e to_reviewer. The Gel Doc system highlights pixels that exceed intensity readings of 65,535 in red, confirming that the bands remained within the quantifiable range (Figure 4c-e to_reviewer).

Figure 4_to_reviewer

Figure 4_to_reviewer. Effect of bafilomycin A1 on B7-H4 expression. Panels a and b show the relative quantification of B7-H4 protein levels in MDA-MB-468 treated with palmostatin B (a) and bafilomycin A1(b). n = 3. Panels c-e show the original Western blot results from Fig. 2k-m. We re-analyzed the Western bands using Image Lab software, confirming that the bands were not saturated.

Reviewer #3 (Remarks to the Author):

The manuscript offers novel insights into the regulation of B7-H4 stability via palmitoylation and its role in tumor immune evasion, with a particular focus on ZDHHC3's involvement in this process. The study is well-designed and highlights an important, though underexplored, post-translational modification with potential therapeutic implications. The authors provide moderate evidence supporting the palmitoylation of B7-H4 at Cys130 and the role of ZDHHC3 in maintaining its stability. Additionally, the CDK4/6 inhibitor abemaciclib is discussed in relation to B7-H4 protein stability; however, it remains unclear whether this effect is dependent on palmitoylation. While the findings are compelling, several key areas require further clarification.

Response:

Thank you for pointing out the novelty and potential therapeutic implications of our findings. Please see our responses to your detailed comments.

Major Comments:

1. In Figures 2g and 2h, the input level of B7-H4 in the 2-BP treated group is noticeably lower compared to the untreated group, which may significantly affect the accurate assessment of palmitoylated B7-H4 levels. Although 2-BP is a broad-spectrum palmitoylation inhibitor, there is insufficient evidence to conclude that it impacts B7-H4 protein stability through palmitoylation.

Response:

The observed decrease in B7-H4 levels following 2-BP treatment likely results from inhibited palmitoylation, leading to downregulation of the B7-H4 protein. In Figures 2g and 2h, we assessed B7-H4 palmitoylation levels in the presence of a palmitoylation inhibitor. While 2-BP broadly inhibits protein palmitoylation, the palmitoylation-depalmitoylation cycle remains dynamic, suggesting that some B7-H4 may still be palmitoylated. Our quantification of B7-H4 palmitoylation in Figures 2g and 2h reveals a significant reduction following 2-BP treatment in both MDA-MB-468 and 4H11 cell lines (see Figure 5_to_reviewer). The quantification data are now included in the revised Figures 2g and 2h. These findings demonstrate that 2-BP effectively suppresses B7-H4 palmitoylation and reduces the overall levels of the B7-H4 protein. We recognize that 2-BP serves as a broad-spectrum palmitoylation inhibitor. As is standard in the field, we utilized it as a tool compound to investigate the potential palmitoylation of B7-H4. Through a series of complementary experiments, we have confirmed and validated that B7-H4 is indeed palmitoylated (see Figures 2 and 3). Our data underscore the critical role of palmitoylation in stabilizing the B7-H4 protein.

Figure 5_to_reviewer

Figure 5_to_reviewer. a, b, Quantification of B7-H4 palmitoylation levels in MDA-MB-468 (a) and 4H11 (b), treated with or without 2-BP (see Fig. 2g,h).

2. In Figure 3a, the input level of B7-H4 in the C130A mutant group is obviously lower relative to the wild-type (WT) group, which raises concerns about concluding that C130 is definitively the palmitoylation site of B7-H4. While the study identifies Cys130 as the palmitoylation site based on mass spectrometry and mutagenesis studies, how certain are we that no other cysteine residues on B7-H4 undergo palmitoylation? Furthermore, how might palmitoylation at other cysteine sites impact the stability or function of B7-H4? Lastly, what is the potential mechanism by which palmitoylation at Cysteine 130 regulates B7-H4 protein stability?

Response:

We are aware that, despite transfecting equal amounts of WT and C130A plasmids into 293T cells, we observed lower expression of C130A B7-H4. This is consistent with our finding that the C130A mutant is less stable than the WT protein (Fig. 3c). We normalized the palmitoylated B7-H4 to the input levels and quantitatively confirmed that palmitoylation of B7-H4 is significantly reduced in the C130A mutant (Figure 6a_to_reviewer).

To further address your comment, we adjusted the amount of B7-H4 (C130A) plasmid during transfection to boost input levels and repeated the ABE assay. Again, C130A manifested a significant reduction in palmitoylation of B7-H4 (Figure 6b_to_reviewer). We have included these data in the revision (Figure 6b_to_reviewer and new Fig. 3a).

Mass spectrometry analysis does not cover every residue. Apart from C130A, we conducted site-directed mutagenesis on other cysteines in B7-H4 and made several B7-H4 mutants. We found no significant impact of other cysteine mutations on the palmitoylation of B7-H4 (Figure 6c_to_reviewer and new Extended Data Fig. 3c). Thus, Cys130 is the primary site of the palmitoylation of B7-H4.

Regarding how palmitoylation at Cys130 regulates B7-H4 stability, it is well-established that palmitoylation functions as a sorting signal to direct proteins to the plasma membrane⁵⁻⁸. Thus, we reason that palmitoylation at Cys130 acts as a sorting signal at the Golgi apparatus, directing B7-H4 to the plasma membrane. Without palmitoylation,

B7-H4 is more likely to be redirected to the lysosome for degradation, preventing its delivery to the plasma membrane, where it would typically inhibit T-cell function.

Figure 6_to_reviewer

Figure 6_to_reviewer. Effect of cysteine residue mutations on the palmitoylation of B7-H4. **a**, B7-H4(C130A) palmitoylation levels were quantified in 293T cells transfected with C130A and WT B7-H4 plasmids (see Fig. 3a). **b**, ABE assay shows the palmitoylation of C130A and WT B7-H4 with equal amounts of B7-H4 input. **c**, ABE assay shows the palmitoylation of WT and different mutated B7-H4 in 293T cells.

3. When using the $\Delta\Delta C_t$ method to calculate fold changes in mRNA expression, it is essential that the primers used for a specific target gene, such as ZDHHC, are designed to amplify the same region of the transcript. If different ZDHHC primers are used, it can result in amplification of different regions of the mRNA, which may not be directly comparable. Differences in primer efficiency, binding affinity, and the region of the gene targeted can lead to variability in C_t values, potentially skewing the results.

Response:

Thank you for raising this methodological comment. We recognize that differences in primer efficiency, binding affinity, and the region of the targeted transcript can introduce variability in C_t values, potentially affecting the accuracy of our results. Our goal is to assess the relative abundance of PATs in tumor cell lines. Given that most PAT antibodies are either unavailable or non-specific, we used qPCR to measure PAT expression and normalized our results to the reference gene ACTB. Notably, we have assessed the PCR primer efficiencies to ensure the accuracy of our ΔC_t calculations. Moreover, we complemented the qPCR data with Acyl-biotin exchange (ABE) assays and evaluated the palmitoylation catalytic capacity of PATs, providing a direct measurement of ZDHHC activity. We have included additional details in the Methods section in the revised manuscript (Page 16, lines 540-544).

4. The study shows that abemaciclib, but not other CDK4/6 inhibitors like palbociclib or ribociclib, promotes B7-H4 lysosomal degradation. What mechanistic explanations

might account for this difference, and how can the role of abemaciclib independent of CDK4/6 inhibition be further validated?

Response:

While CDK4/6 inhibitors, including abemaciclib and palbociclib, were originally developed to target CDK4/6 specifically, recent studies have uncovered significant differences in their molecular signatures, pharmacokinetics, and clinical toxicities^{9,10}. Specifically, Hafner et al. have reported that abemaciclib treatment induces a distinct transcriptional signature, which is absent or weakly present in breast cancer cell lines treated with palbociclib⁹. Our analysis of their RNA-seq data, combined with GO enrichment analysis, indicates that protein catabolic processes and cellular component biogenesis (including lysosomal biogenesis) are significantly upregulated in abemaciclib-treated cells compared to palbociclib-treated cells (Figure 7a_to_reviewer). Additionally, using isobaric tandem mass tag (TMT) MS for phosphoproteome profiling, Hafner et al. have observed different kinase activities in response to abemaciclib versus palbociclib treatment⁹. When normalizing the Log2FC changes of abemaciclib's targets to palbociclib's targets, we found that kinases involved in the catabolic process and Golgi-related vesicle trafficking were enriched following abemaciclib treatment (Figure 7b_to_reviewer). In contrast, DNA and RNA-related processes were more prominently enriched in palbociclib-treated cells (Figure 7c_to_reviewer). Despite the poorly defined mechanistic details, these results support the notion that abemaciclib and palbociclib may be biologically distinct.

To define and validate that the effect of abemaciclib on B7-H4 is independent of CDK4/6 inhibition, we first demonstrated that abemaciclib downregulates B7-H4 in MDA-MB-468 cells, which are Rb-deficient¹¹. This suggests that abemaciclib's effect on B7-H4 can be independent of the CDK4/6-Rb pathway. Then, we showed that cell cycle arrest induced by nocodazole did not overlap with the impact of abemaciclib on B7-H4 (Fig. 4l and Extended Data Fig. 4n), further supporting the possibility that abemaciclib regulates B7-H4 degradation independently of cell cycle arrest. Altogether, these findings indicate that abemaciclib's effect on B7-H4 can be different from its inhibitory role in CDK4/6.

[Figure redacted]

Figure 7_to_reviewer. Different roles of abemaciclib and paboliclib in breast cancer cell lines⁹. Go enrichment analysis of RNA-seq data (Table S1 from Reference 9) (a) and go enrichment analysis of phosphoproteome data (Table S2 from Reference 9) (b, c) from cells treated with abemaciclib or paboliclib.

5. According to the paper, abemaciclib promotes the lysosomal degradation of B7-H4 but does not directly influence its palmitoylation. Rather, the study demonstrates that abemaciclib enhances lysosomal biogenesis and increases lysosomal capacity, leading to the degradation of B7-H4 in cancer cells. The palmitoylation of B7-H4 plays a crucial role in stabilizing the protein by protecting it from lysosomal degradation. Abemaciclib, by promoting lysosomal activity, counteracts this stabilization, ultimately reducing B7-H4 levels despite its palmitoylation.

Response:

As you rightly pointed out, abemaciclib indirectly reduces B7-H4 stability by enhancing lysosomal degradation. While palmitoylation serves to stabilize the B7-H4 protein by inhibiting its lysosomal degradation, abemaciclib promotes lysosomal activity, effectively counteracting this stabilization. As a result, B7-H4 levels decrease despite its palmitoylation.

6. The paper suggests repurposing abemaciclib to treat B7-H4+ tumors based on preclinical models. What would be the major challenges in translating this finding to clinical trials, and how might patient-specific factors, such as variations in B7-H4 expression or lysosomal activity, impact the efficacy of this approach?

Response:

We agree with you that patient heterogeneity poses challenges to clinical translation of our preclinical findings. While B7-H4 is generally highly expressed in breast cancer, its expression varies across individuals, and such variability could impact the effectiveness of abemaciclib in inhibiting B7-H4-mediated immune suppression. Clinical sequencing or immunohistochemistry to assess B7-H4 expression in tumor samples would be necessary for selecting appropriate patients for treatment. Furthermore, differences in lysosomal activity and biogenesis across tumor types and among patients could also impact the efficacy of abemaciclib. Identifying and validating effective lysosomal biomarkers that correlate with clinical outcomes following abemaciclib treatment will be essential. Establishing these biomarkers could help refine patient selection and enhance the precision of this therapeutic approach.

We anticipate that the efficacy of abemaciclib would be enhanced in patients with relatively high B7-H4 expression and/or active lysosomal biogenesis. We propose stratifying patients based on B7-H4 expression levels and lysosomal biomarkers to optimize treatment strategies. We have expanded the discussion section covering these potential strategies for clinical translation (Page 10, lines 308-310).

7. Abemaciclib demonstrates strong efficacy in vivo by reducing tumor burden. How can the authors be convinced that this tumor growth attenuation occurs independently of CDK4/6 inhibition?

Response:

As a CDK4/6 inhibitor, abemaciclib is known to impede tumor progression through its action on CDK4/6. However, our research reveals that abemaciclib can also downregulate B7-H4 independently of CDK4/6 in vitro (Fig. 4j,k and Extended Data Fig. 4i,j), suggesting that this mechanism may contribute to its anti-tumor efficacy in vivo. In both the PyMT and CT26 tumor models, we found that abemaciclib treatment led to decreased B7-H4 expression in tumors and inhibited tumor growth (see Fig. 5a-e; new Extended Data Fig. 5c,d, and f-h; Figure 1_to_reviewer). Additionally, the anti-tumor effect of abemaciclib was more pronounced in B7-H4⁺ CT26 tumors compared to B7-H4⁻ CT26 tumors (new Extended Data Fig. 5e-i). These findings suggest that abemaciclib can inhibit tumor progression in a manner dependent on B7-H4 and distinct from its inhibitory role in CDK4/6.

Reviewer #4 (Remarks to the Author):

B7-H4, a ligand from the B7 family with immunosuppressive functions, is identified as a potential target for cancer immunotherapy. In this study, Yan et al. found that ZDHHC3, a zinc finger DHHC-type palmitoyltransferase, catalyzes the palmitoylation of B7-H4 at Cys130. This modification prevents lysosomal degradation, thereby maintaining B7-H4 stability and enhancing its immunosuppressive effects, which promotes tumor growth. The researchers demonstrated that abemaciclib, a CDK4/6 inhibitor, could be repurposed to degrade B7-H4 by activating lysosomes, independently of its effects on the cell cycle. This action promotes T cell activation and counteracts B7-H4-mediated immune suppression.

This is an elegant study that is robustly designed, well conducted, and clearly written, providing valuable insights to enhance our understanding of PTM in immuno-oncology. This reviewer only has a few comments that require some clarification.

Response:

We are grateful for your positive assessment of our study. Please see our point-by-point responses to your comments.

(1) The authors convincingly demonstrated the antitumor immune activation resulting from targeting the palmitoylation of B7-H4 in tumor cells. Are there other cells, such as macrophages or neutrophils, within the TME that express B7-H4? Would targeting ZDHHC3 or abemaciclib reduce B7-H4 levels on those cells?

Response:

Single-cell RNA-seq data from TISCH2 showed that breast cancer cells expressed the highest levels of B7-H4 in the tumor microenvironment (Figure 8_to_reviewer). Moreover, it is technically challenging to isolate sufficient primary tumor-associated macrophages (TAMs) and neutrophils from fresh and limited breast cancer tissue specimens for genetic manipulation of ZDHHC3 and B7-H4. We have included this point in the revised manuscript (Page 9, lines 277-279).

Figure 8_to_reviewer

Figure 8_to_reviewer. Gene expression of *VTCN1* in different cell subsets of breast cancers (TISCH2, BRCA datasets).

(2) B7-H4 palmitoylation is a novel finding. It is somewhat unclear how generalizable this is. Are there any data from patient samples to support this?

Response:

We have demonstrated that B7-H4 manifests palmitoylation in multiple human and mouse breast cancer cell lines. In response to your inquiry about patient samples, we detected B7-H4 palmitoylation in cancer samples from patients with ovarian cancer. We have included this finding in the revision (new Extended Data Fig. 2k).

References

1. Song, X. *et al.* Pharmacologic Suppression of B7-H4 Glycosylation Restores Antitumor Immunity in Immune-Cold Breast Cancers. *Cancer Discov.* **10**, 1872–1893 (2020).
2. S. Mesquita, F. *et al.* Mechanisms and functions of protein S-acylation. *Nat. Rev. Mol. Cell Biol.* **25**, 488–509 (2024).
3. Rana, M. S. *et al.* Fatty acyl recognition and transfer by an integral membrane S-acyltransferase. *Science* **359**, eaao6326 (2018).
4. Resh, M. D. Open Biology: overview for special issue on dynamics of protein fatty acylation. *Open Biol.* **11**, 210228 (2021).
5. Wang, J. *et al.* ARF6 plays a general role in targeting palmitoylated proteins from the Golgi to the plasma membrane. *J. Cell Sci.* **136**, jcs261319 (2023).
6. Guo, H. *et al.* Targeting EGFR-dependent tumors by disrupting an ARF6-mediated sorting system. *Nat. Commun.* **13**, 6004 (2022).
7. Ford, C., Parchure, A., von Blume, J. & Burd, C. G. Cargo sorting at the trans-Golgi network at a glance. *J. Cell Sci.* **134**, jcs259110 (2021).
8. Ernst, A. M. *et al.* S-Palmitoylation Sorts Membrane Cargo for Anterograde Transport in the Golgi. *Dev. Cell* **47**, 479-493.e7 (2018).
9. Hafner, M. *et al.* Multiomics Profiling Establishes the Polypharmacology of FDA-Approved CDK4/6 Inhibitors and the Potential for Differential Clinical Activity. *Cell Chem. Biol.* **26**, 1067-1080.e8 (2019).
10. Klein, M. E., Kovatcheva, M., Davis, L. E., Tap, W. D. & Koff, A. CDK4/6 inhibitors: The mechanism of action may not be as simple as once thought. *Cancer Cell* **34**, 9–20 (2018).
11. Robinson, T. J. W. *et al.* RB1 Status in Triple Negative Breast Cancer Cells Dictates Response to Radiation Treatment and Selective Therapeutic Drugs. *PLOS ONE* **8**, e78641 (2013).

Reviewer #1 (Remarks to the Author):

The authors have adequately addressed my comments in the revised manuscript. I have no further concerns. Congratulations to the research team!

Reviewer #2 (Remarks to the Author):

In the revised manuscript the authors addressed all concerns requested by this reviewer and also added some additional experiments, which clarified and reinforced their observations.

Reviewer #3 (Remarks to the Author):

The revised manuscript is meticulously organized, and all concerns raised by the reviewers have been thoroughly and effectively addressed.

Reviewer #4 (Remarks to the Author):

The revised manuscript addressed my previous concerns. I have no further comments.

Response:

We sincerely appreciate all reviewers for their constructive feedback and for dedicating time to evaluate our revised manuscript. We're grateful for the positive remarks and are thankful that our revisions have met your expectations.